# Ringleader ASGD: The First Asynchronous SGD with Optimal Time Complexity under Data Heterogeneity

**Artavazd Maranjyan, Peter Richtárik**
King Abdullah University of Science and Technology (KAUST)
{arto.maranjyan,richtarik}@gmail.com

## Abstract

Asynchronous stochastic gradient methods are central to scalable distributed optimization, particularly when devices differ in computational capabilities. Such settings arise naturally in federated learning, where training takes place on smartphones and other heterogeneous edge devices. In addition to varying computation speeds, these devices often hold data from different distributions. However, existing asynchronous SGD methods struggle in such heterogeneous settings and face two key limitations. First, many rely on unrealistic assumptions of similarity across workers' data distributions. Second, methods that relax this assumption still fail to achieve theoretically optimal performance under heterogeneous computation times. We introduce Ringleader ASGD, the first asynchronous SGD algorithm that attains the theoretical lower bounds for parallel first-order stochastic methods in the smooth nonconvex regime, thereby achieving optimal time complexity under data heterogeneity and without restrictive similarity assumptions. Our analysis further establishes that Ringleader ASGD remains optimal under arbitrary and even time-varying worker computation speeds, closing a fundamental gap in the theory of asynchronous optimization.

## 1 Introduction

Modern machine learning increasingly depends on large-scale distributed training across clusters with hundreds or even thousands of GPUs (Shoeybi et al., 2019; Brown et al., 2020; Narayanan et al., 2021). However, classical synchronous training methods struggle to scale, as device failures, network instabilities, and synchronization overheads introduce inefficiencies (Chen et al., 2016; Grattafiori et al., 2024). These issues become even more pronounced in environments with heterogeneous computational power, such as Federated Learning (FL), where devices range from high-end datacenter GPUs to resource-constrained edge hardware (Konečný et al., 2016; McMahan et al., 2016; Li et al., 2020; Kairouz et al., 2021). Because synchronous methods are bottlenecked by the slowest participants, faster devices remain idle, leading to underutilization of resources when stragglers—nodes slowed by computation or communication—lag significantly behind.

One way to reduce synchronization bottlenecks is to equip data centers with homogeneous GPUs. However, this approach is expensive and difficult to scale: upgrading to faster GPUs requires replacing all devices simultaneously, since heterogeneous hardware cannot be combined efficiently. Even then, homogeneity does not eliminate synchronization issues, as hardware failures and device dropouts still cause stragglers and idle time. Moreover, this solution applies only to controlled datacenter environments and is infeasible in FL, where edge devices are outside the server's control.

A more promising approach is to shift from hardware to algorithmic solutions using asynchronous optimization methods. These methods remove the need for synchronization, allowing fast workers to contribute without waiting for slower ones (Tsitsiklis et al., 1986; Recht et al., 2011; Agarwal & Duchi, 2011; Dean et al., 2012; Li et al., 2014). Despite their appeal, asynchronous methods are harder to analyze. In particular, a meaningful analysis would require studying *time to convergence*, rather than iteration complexity only. Iteration count alone does not reflect training speed in parallel settings: a method with more iterations may still finish faster if those steps avoid waiting for slow

Table 1: Comparison of time complexities for parallel first-order methods under the fixed computation time model, where worker $i$ needs $\tau_i$ seconds to compute a stochastic gradient, with the times ordered so that $\tau_n$ is the largest (2). We denote by $\tau_{\text{avg}} := \frac{1}{n} \sum_{i=1}^{n} \tau_i$ the average computation time. Problem parameters include the initial function suboptimality $\Delta := f(x^0) - f^*$ (Assumption 3), the target stationarity $\varepsilon$, the variance bound of the stochastic gradients $\sigma^2$ (Assumption 1), and smoothness constants. Specifically, $L_f$ is the smoothness constant of $f$ (3); $L_{\max} := \max_{i \in [n]} L_{f_i}$ with $L_{f_i}$ the smoothness constant of $f_i$; and $L$ is a constant associated with our new smoothness-type assumption (Assumption 2). They satisfy $L_f \leq L \leq L_{\max}$ (Lemma E.1). All stated time complexities hide universal constant factors. Each column indicates whether a method satisfies the following desirable properties: **Optimal:** achieves the theoretical lower bound derived by Tyurin & Richtárik (2024) for parallel first-order stochastic methods in heterogeneous data setting. **No sync.:** does not require synchronization and is therefore *asynchronous*. **No idle workers:** all workers remain busy without waiting, so computational resources are fully utilized. **No discarded work:** no computation is wasted, and no worker is stopped mid-computation.

| Method | Time Complexity | Optimal | No sync. | No idle workers | No discarded work |
|---|---|---|---|---|---|
| Naive Minibatch SGD (Section 3) | $\frac{L_f \Delta}{\varepsilon}\left(\tau_n + \tau_n \frac{\sigma^2}{n\varepsilon}\right)$ | ✘ | ✘ | ✘ | ✔ |
| IA²SGD (Wang et al., 2025) (Appendix H) | $\frac{L_{\max}\Delta}{\varepsilon}\left(\tau_n + \tau_n \frac{\sigma^2}{n\varepsilon}\right)$ [†] | ✘ | ✔ | ✔ | ✔ |
| Malenia SGD (Tyurin & Richtárik, 2024) | $\frac{L_f \Delta}{\varepsilon}\left(\tau_n + \tau_{\text{avg}} \frac{\sigma^2}{n\varepsilon}\right)$ | ✔ | ✘ | ✔ | ✘ |
| Ringleader ASGD **(new)** (Algorithm 1; Theorem 5.3) | $\frac{L\Delta}{\varepsilon}\left(\tau_n + \tau_{\text{avg}} \frac{\sigma^2}{n\varepsilon}\right)$ | ✔[‡] | ✔ | ✔ | ✔ |
| Lower Bound (Tyurin & Richtárik, 2024) | $\frac{L_f \Delta}{\varepsilon}\left(\tau_n + \tau_{\text{avg}} \frac{\sigma^2}{n\varepsilon}\right)$ | — | — | — | — |

[†] The analysis of Wang et al. (2025) assumes that all $f_i$ share the same smoothness constant, i.e., each $f_i$ is $L_{f_i}$–smooth and the bound $L_{\max}$ is used for all. This assumption is unnecessary: under our Assumption 2, the constant improves to $L$ while the analysis remains unchanged. [‡] The time complexities of Ringleader ASGD and Malenia SGD differ in the smoothness constant only. Since Malenia SGD is optimal, Ringleader ASGD is also optimal whenever $L$ exceeds $L_f$ by at most a universal constant factor, that is, $L = \mathcal{O}(L_f)$.

workers. This raises a fundamental question: *among all parallel methods, which ones are provably fastest in theory?* To make this precise, we restrict our attention to smooth nonconvex problems and to stochastic first-order methods, encompassing algorithms with or without synchronization.

Recently, Tyurin & Richtárik (2024) studied this regime, where they derived lower bounds. They then proposed two algorithms: Rennala SGD, designed for the *homogeneous data setting*, where all workers draw samples from the same distribution, and Malenia SGD, for the *heterogeneous data setting*, where data distributions differ across workers. They showed that both methods are optimal—achieving the lower bounds—and, perhaps surprisingly, both are synchronous (they periodically synchronize workers). The key idea in both is to fully utilize the available computational resources by keeping workers continuously busy: each worker computes independently, and synchronization occurs only after a sufficient number of gradient computations have been accumulated.

At first, the result of Tyurin & Richtárik (2024) suggested a rather pessimistic outlook for asynchronous methods: despite their practical appeal, they showed that existing asynchronous methods are not optimal and that the method achieving the lower bound is *synchronous*. This created the view that optimality is inherently tied to synchronization. However, this view was overturned by Maranjyan et al. (2025d), who, in the *homogeneous data setting*, introduced Ringmaster ASGD—the first asynchronous SGD method to achieve optimal time complexity as the synchronous Rennala SGD. Although both methods share the same theoretical guarantees, Ringmaster ASGD can be faster than Rennala SGD in practice, since it avoids synchronization and benefits from more frequent updates.

Nevertheless, the work of Maranjyan et al. (2025d) established optimality in the *homogeneous data setting* only. The question of whether some variant of a parallel method that does not rely on synchronization (i.e., is asynchronous) can also be optimal in the more general *heterogeneous data setting* remained open. In this work, we close this gap and answer the question affirmatively.

The relevance of the heterogeneous data setting is clear in FL, where participants naturally hold distinct datasets (Zhao et al., 2018; Li et al., 2020; Tan et al., 2022). This setting is more chal-

lenging than the homogeneous one, since the usual asynchronous SGD philosophy—updating after every gradient—can bias training toward fast workers' local data. Most existing methods address this by assuming similarity across client distributions (Mishchenko et al., 2022; Koloskova et al., 2022; Nguyen et al., 2022; Islamov et al., 2024), an assumption that simplifies analysis but is often unrealistic in practice (e.g., hospitals with distinct demographics, mobile users in different regions).

Recent work by Wang et al. (2025) took a step toward removing these restrictive assumptions by proposing Incremental Aggregated Asynchronous SGD (IA²SGD), a method that provably converges without similarity assumptions. However, their method achieves the same time complexity as standard Naive Minibatch SGD (see the first two rows of Table 1)—the simplest synchronous SGD baseline, which waits to collect one gradient from each worker before every update—thus failing to provide the computational advantages that motivate asynchronous approaches in the first place.

To the best of our knowledge, the only method proven to be optimal in the *heterogeneous data setting* is the synchronous algorithm Malenia SGD of Tyurin & Richtárik (2024), which, notably, does not rely on similarity assumptions. However, synchronization is a major bottleneck in practice: although synchronous and asynchronous methods can share the same theoretical complexity, asynchronous methods are often faster in practice because they avoid costly synchronization and benefit from more frequent updates, as demonstrated in the homogeneous case by Maranjyan et al. (2025d).

This raises a fundamental question: *Is it possible to design an asynchronous method that requires no similarity assumptions while still achieving optimal time complexity?* In this paper, we answer this question affirmatively by introducing Ringleader ASGD, the first asynchronous SGD method that achieves optimal time complexity[1] in the *heterogeneous data setting*. Importantly, Ringleader ASGD attains this without relying on restrictive similarity assumptions.

## 1.1 CONTRIBUTIONS

Our main contributions are the following:

**Optimal asynchronous SGD under data heterogeneity.** We prove that Ringleader ASGD (Algorithm 1) is, to the best of our knowledge, the first asynchronous method in the heterogeneous setting under the fixed computation model (2) matching the lower bounds for parallel methods of Tyurin & Richtárik (2024), when the smoothness-type constant $L$ in Assumption 2 is within a constant factor of the smoothness $L_f$ used to obtain the lower bounds (Table 1). Importantly, Ringleader ASGD attains this without any similarity assumptions across clients' data.

**Additional useful properties.** Beyond optimal time complexity, Ringleader ASGD satisfies two additional properties: (i) all workers remain continuously active (*no idle workers*), and (ii) every computed gradient is used in the update (*no discarded work*). These properties are crucial in practice, ensuring full resource utilization: all workers contribute at all times, and no computation is wasted. Table 1 compares Ringleader ASGD with benchmark algorithms on these properties.

**Parameter-free design.** Unlike the optimal synchronous method Malenia SGD (Tyurin & Richtárik, 2024), which requires prior knowledge of the gradient variance bound and target accuracy, our method operates in the fixed computation time model without such parameters (except for the stepsize, needed only to match the optimal rate). This makes it more practical for real-world deployments, where these quantities are usually unknown or hard to estimate. The same parameter-free improvement can also be applied to Malenia SGD, as we discuss in Appendix I.

**Universal computation model.** In Appendix D, we extend our analysis beyond the fixed computation time model to the setting of arbitrarily varying times, accommodating virtually any computational behavior, including stochastic or adversarial patterns, while retaining optimal time complexity.

**Empirical validation.** In Appendix A, we evaluate Ringleader ASGD on toy problems, confirming the theory and showing clear practical gains over baselines.

---

[1] Throughout the paper, we refer to our method as optimal. Formally, this holds whenever the constant $L$—associated with our new smoothness-type assumption (Assumption 2)—is at most a constant factor larger than the smoothness constant $L_f$ in the derived lower bounds (Tyurin & Richtárik, 2024). See Table 1 for details.

## 2    PROBLEM SETUP

We consider a distributed learning setting with $n$ workers, where each worker $i$ possesses its own local data distribution $\mathcal{D}_i$. Our goal is to solve the following distributed optimization problem:

$$\operatorname*{minimize}_{x \in \mathbb{R}^d} \left\{ f(x) := \frac{1}{n} \sum_{i=1}^{n} f_i(x) \right\}, \quad \text{where} \quad f_i(x) := \mathbb{E}_{\xi_i \sim \mathcal{D}_i} \left[ f_i(x; \xi_i) \right] . \tag{1}$$

Here $f_i \colon \mathbb{R}^d \to \mathbb{R}$ denotes the local objective of worker $i$, defined as the expectation of the sample loss $f_i(x; \xi_i)$ over data points $\xi_i$ drawn from its local distribution $\mathcal{D}_i$.

We first focus on the case where workers have constant computation speeds, as this setting is more intuitive and serves as a foundational model for understanding the dynamics of asynchronous distributed optimization. The extension to arbitrary computation times is presented in Appendix D.

Following the *fixed computation model* (Mishchenko et al., 2022), we formalize:

> Each worker $i$ requires $\tau_i$ seconds[2] to compute one stochastic gradient $\nabla f_i(x, \xi_i)$ .
>
> Without loss of generality, assume $0 < \tau_1 \leq \tau_2 \leq \cdots \leq \tau_n$ . $\tag{2}$

 We assume communication is infinitely fast (taking 0 seconds), both from workers to the server and from the server to workers[3]. This is a modeling choice—arguably the simplest—and has been standard in prior work (Mishchenko et al., 2022; Koloskova et al., 2022; Tyurin & Richtárik, 2024; Maranjyan et al., 2025d), even if not always stated explicitly. We further discuss the motivation and limitations of this abstraction in Appendix C. A related study by Tyurin et al. (2024b) considers the case where communication is non-negligible and proposes techniques to address it.

Finally, we denote by $\tau_{\text{avg}} := \frac{1}{n} \sum_{i=1}^{n} \tau_i$ the average computation time across all workers.

### 2.1    NOTATIONS

We denote the standard inner product in $\mathbb{R}^d$ by

$$\langle x, y \rangle = \sum_{i=1}^{d} x_i y_i ,$$

and the corresponding Euclidean norm by $\|x\| := \sqrt{\langle x, x \rangle}$. We use $[n] := \{1, 2, \ldots, n\}$ to denote the index set, and $\mathbb{E}\left[\cdot\right]$ for mathematical expectation. For functions $\phi, \psi : \mathcal{Z} \to \mathbb{R}$, we write $\phi = \mathcal{O}(\psi)$ if there exists a constant $C > 0$ such that $\phi(z) \leq C\psi(z)$ for all $z \in \mathcal{Z}$.

### 2.2    ASSUMPTIONS

We consider the following standard assumptions for the nonconvex setting.

**Assumption 1.** *For each $i \in [n]$ and every $\xi$, the function $f_i(x; \xi)$ is differentiable with respect to its first argument $x$. Moreover, the stochastic gradients are unbiased and have bounded variance $\sigma^2 \geq 0$, that is,*

$$\mathbb{E}_{\xi_i \sim \mathcal{D}_i} \left[ \nabla f_i(x; \xi_i) \right] = \nabla f_i(x), \quad \forall x \in \mathbb{R}^d, \ \forall i \in [n] ,$$

$$\mathbb{E}_{\xi_i \sim \mathcal{D}_i} \left[ \|\nabla f_i(x; \xi_i) - \nabla f_i(x)\|^2 \right] \leq \sigma^2, \quad \forall x \in \mathbb{R}^d, \ \forall i \in [n] .$$

**Assumption 2.** *Each function $f_i$ is differentiable. There exists a constant $L > 0$ such that, for all $x \in \mathbb{R}^d$ and $y_1, \ldots, y_n \in \mathbb{R}^d$,*

$$\left\| \nabla f(x) - \frac{1}{n} \sum_{i=1}^{n} \nabla f_i(y_i) \right\|^2 \leq \frac{L^2}{n} \sum_{i=1}^{n} \|x - y_i\|^2 .$$

---

[2]One could alternatively assume that each worker requires *at most* $\tau_i$ seconds. Under this formulation, all of our upper bounds would still hold; however, the lower bound would no longer be valid. For this reason, we adopt the assumption that each worker requires exactly $\tau_i$ seconds.

[3]Alternatively, one could define $\tau_i$ as the time required for a worker to both compute a gradient and communicate it to the server, while keeping server-to-worker communication infinitely fast. Our upper bounds would still hold under this formulation, but the lower bounds would no longer apply, so we use the simpler model.

Recall that a differentiable function $\phi \colon \mathbb{R}^d \to \mathbb{R}$ is called $L_\phi$–smooth if

$$\|\nabla \phi(x) - \nabla \phi(y)\| \leq L_\phi \|x - y\|, \quad \forall x, y \in \mathbb{R}^d . \tag{3}$$

By convention, $L_\phi$ denotes the smallest such constant.

Note that Assumption 2 is stronger than requiring $f$ itself to be $L_f$–smooth, yet weaker than all $f_i$ being $L_{f_i}$–smooth. The constants satisfy the following relation (Lemma E.1)

$$L_f \ \leq \ L \ \leq \ \sqrt{\frac{1}{n} \sum_{i=1}^{n} L_{f_i}^2} \ \leq \ \max_{i \in [n]} L_{f_i} =: L_{\max} .$$

To the best of our knowledge, prior work on asynchronous SGD under data heterogeneity has always assumed smoothness of each $f_i$ (Koloskova et al., 2022; Nguyen et al., 2022; Wang et al., 2025).

**Assumption 3.** *There exists $f^* > -\infty$ such that $f(x) \geq f^*$ for all $x \in \mathbb{R}^d$. We define $\Delta := f(x^0) - f^*$, where $x^0$ is the starting point of the optimization methods.*

We seek an $\varepsilon$–stationary point—a (possibly random) vector $x$ satisfying $\mathbb{E}\left[\|\nabla f(x)\|^2\right] \leq \varepsilon$.

## 3 BACKGROUND AND MOTIVATION

In this section, we review relevant existing methods in distributed optimization and discuss their limitations to motivate our algorithmic design.

**Naive Minibatch SGD.** This algorithm is the most straightforward way to solve (1) in a distributed setting. At each iteration $k$, the server waits for one gradient from every worker and updates

$$x^{k+1} = x^k - \gamma \frac{1}{n} \sum_{i=1}^{n} \nabla f_i \left( x^k; \xi_i^k \right) .$$

This guarantees an unbiased minibatch gradient but synchrony makes it inefficient: the iteration time is set by the slowest worker, $\tau_n = \max_i \tau_i$ (2), so fast workers idle while waiting for stragglers.

**Malenia SGD.** The algorithm (Tyurin & Richtárik, 2024) addresses straggler problem by keeping all workers active: each computes gradients at $x^k$ while the server accumulates them until

$$\left( \frac{1}{n} \sum_{i=1}^{n} \frac{1}{b_i^k} \right)^{-1} \geq \max \left\{ 1, \frac{\sigma^2}{n\varepsilon} \right\} \tag{4}$$

where $b_i^k$ is the number of gradients received from worker $i$. Once this holds, the update

$$x^{k+1} = x^k - \gamma \bar{g}^k, \qquad \bar{g}^k := \frac{1}{n} \sum_{i=1}^{n} \bar{g}_i^k, \ \ \bar{g}_i^k := \frac{1}{b_i^k} \sum_{j=1}^{b_i^k} \nabla f_i \left( x^k; \xi_i^{k,j} \right)$$

is performed and broadcast synchronously. This achieves the optimal time complexity, but requires knowledge of $\sigma$ and $\varepsilon$ and discards ongoing computations during synchronization.

**Toward Asynchronous Methods.** A naive way to make optimization asynchronous is to update immediately upon receiving a gradient:

$$x^{k+1} = x^k - \gamma \nabla f_{i^k} \left( x^{k-\delta^k}; \xi_{i^k}^{k-\delta^k} \right) ,$$

where $i^k$ denotes the worker that sent the gradient at iteration $k$, and $\delta^k \geq 0$ is its delay, i.e., the number of server updates that occurred while the gradient was being computed. Delays arise naturally: fast workers return gradients quickly, while slow ones compute on stale iterates.

This scheme is biased under heterogeneous data. Fast workers dominate updates, pulling the model toward their local minima, while slow workers contribute too rarely, so iterates may oscillate or

stagnate. Such bias also blocks classical SGD-style proofs, which rely on one-step progress toward minimizing the global function, but here each update reflects a different local objective. Without data similarity assumptions (Mishchenko et al., 2022; Koloskova et al., 2022; Nguyen et al., 2022; Islamov et al., 2024), it is harder to extend the analysis to the global function—yet such assumptions are rarely realistic when data can be arbitrarily heterogeneous across machines or organizations.

The root issue is that each update uses only one worker's gradient. A better approach is to aggregate information from all workers, even if some gradients are stale, as in Incremental Aggregated Gradient (IAG) (Blatt et al., 2007; Gurbuzbalaban et al., 2017; Vanli et al., 2018) or Stochastic Averaged Gradient (SAG) (Roux et al., 2012; Schmidt et al., 2017).

**IA²SGD.** This method (Wang et al., 2025) constructs updates using information from all workers by maintaining a gradient table on the server, similar to SAG or IAG but with asynchronous table updates. The table is initialized with one stochastic gradient from each worker at $x^0$, after which the server updates

$$x^{k+1} = x^k - \gamma \bar{g}, \qquad \bar{g} = \frac{1}{n} \sum_{i=1}^{n} g_i ,$$

where $g_i$ is the most recent stochastic gradient in the table from worker $i$ (possibly evaluated at a stale iterate $x^{k-\delta_i^k}$). Each arrival replaces the corresponding entry and triggers an update, so every step uses the latest information from all workers while avoiding global synchronization.

The iteration complexity of the algorithm was established by Wang et al. (2025); converting it to the fixed computation time model (see Appendix H) yields the same time complexity as Naive Minibatch SGD (Table 1). Thus, asynchronous execution alone does not yield speedups.

**Motivation.** The main limitation of asynchronous algorithms is gradient delay, which can severely harm convergence by pushing steps along suboptimal trajectories. This issue arises even in homogeneous settings ($f_i \equiv f$). The state-of-the-art solution, Ringmaster ASGD (Maranjyan et al., 2025d), achieves optimal time by discarding overly delayed gradients.

In heterogeneous settings, however, this strategy fails. Slow workers inevitably suffer large delays; if their gradients are discarded, their table entries remain outdated and may never refresh, creating a persistent information bottleneck that harms convergence. One option is to drop some updates from fast workers to balance delays, but this contradicts the core principle of asynchronous methods, where all workers compute continuously.

Instead, our approach *buffers* gradients from fast workers rather than applying them immediately. By updating only once enough gradients are collected, we control delays while keeping all workers active. Buffering also offers a second benefit: aggregating multiple gradients from the same iterate reduces variance, further improving convergence.

## 4 RINGLEADER ASGD

Ringleader ASGD (Algorithm 1) builds on three key insights from prior work. First, inspired by IA²SGD, we maintain a gradient table to ensure information from all workers is incorporated into every update, eliminating the need for data similarity assumptions between workers. Second, following Ringmaster ASGD, we control gradient delays for efficient asynchronous optimization. Third, drawing from Malenia SGD, we buffer rather than discard delayed ones to control delays while preserving valuable computations, enabling continuous utilization of all resources.

An important property of the algorithm is that all workers remain continuously active. Each worker sends a gradient to the server immediately after computing it, with communication assumed instantaneous (Section 2). On receipt, the server either buffers the gradient—in which case the worker simply continues computing and sending new ones—or applies it to update the model. If an update occurs, the new model is instantly returned to the worker, which resumes at the new point. Thus, workers are never idle.

The algorithm proceeds in rounds of exactly $n$ model updates—one per worker. When a worker sends a stochastic gradient, the server may apply an update and return the updated model to that

---

**Algorithm 1** Ringleader ASGD (server algorithm)

---

1: **Input:** Stepsize $\gamma > 0$, initial point $x^0 \in \mathbb{R}^d$
2: **Initialization:** Broadcast $x^0$ to all workers, which then start running Algorithm 2 in parallel
3: Set $k = 0$, $S = \emptyset$; initialize $G_i = 0$, $b_i = 0$ for all $i \in [n]$
4: **while** True **do**
5:      — **Phase 1: await stochastic gradients from all workers** —
6:      **while** $S \neq [n]$ **do**
7:          Receive stochastic gradient $g_j^k$ (computed at $x^{k-\delta_j^k}$) from some worker $j \in [n]$
8:          $G_j = G_j + g_j^k$;   $b_j = b_j + 1$;   $S = S \cup \{j\}$
9:      **end while**
10:      — **Phase 2: perform exactly one update for every worker** —
11:      $x^{k+1} = x^k - \gamma \frac{1}{n} \sum_{i=1}^n G_i / b_i$          $\diamond$ Update using averaged gradients from all workers
12:      Broadcast $x^{k+1}$ to worker $j$                           $\diamond$ Last worker to complete Phase 1
13:      $k = k + 1$;   $S = S \setminus \{j\}$
14:      $g_i^+ = 0, b_i^+ = 0$ for all $i \in [n]$;   $S^+ = \emptyset$     $\diamond$ Initialize temporary buffer for the next round
15:      **while** $S \neq \emptyset$ **do**
16:          Receive stochastic gradient $g_j^k$ $\left(\text{computed at } x^{k-\delta_j^k}\right)$ from some worker $j \in [n]$
17:          **if** $j \in S$ **then**
18:              $G_j = G_j + g_j^k$;   $b_j = b_j + 1$
19:              $x^{k+1} = x^k - \gamma \frac{1}{n} \sum_{i=1}^n G_i / b_i$
20:              Broadcast $x^{k+1}$ to worker $j$
21:              $k = k + 1$;   $S = S \setminus \{j\}$
22:          **else**
23:              $G_j^+ = G_j^+ + g_j^k$;   $b_j^+ = b_j^+ + 1$;   $S^+ = S^+ \cup \{j\}$         $\diamond$ Buffer for next round
24:          **end if**
25:      **end while**
26:      $G_i = G_i^+$;   $b_i = b_i^+$ for all $i \in [n]$;   $S = S^+$     $\diamond$ Transfer buffered gradients to main table
27: **end while**

---

**Algorithm 2** Worker $i$'s subroutine

---

1: **Input:** Model $x$
2: **while** True **do**
3:      Compute $g_i = \nabla f_i(x; \xi_i)$ using a freshly sampled data point $\xi_i \sim \mathcal{D}_i$
4:      Send $g_i$ to the server
5: **end while**

---

worker, but it ensures that each worker receives an updated model at most once per round. Repeating this $n$ times guarantees every worker obtains exactly one new model per round, which keeps delays bounded. To avoid discarding computations from fast workers, their extra gradients are buffered and applied later at the appropriate time, ensuring each round still consists of exactly $n$ updates.

Each round has two *phases*: first, stochastic gradients are buffered until at least one from every worker is available; then exactly $n$ updates are performed (one per worker), after which the old gradients are discarded and the next round begins.

## 4.1 DETAILED DESCRIPTION

**Initialization.** The server broadcasts $x^0 \in \mathbb{R}^d$ to all workers, which then start the worker subroutine (Algorithm 2). Each worker continuously computes stochastic gradients and sends them to the server; if not updated, it simply continues, so workers are never idle.

The server maintains a gradient table $\{(G_i, b_i)\}_{i=1}^n$, initialized with $G_i = 0$ and $b_i = 0$. Here $G_i$ stores accumulated gradients and $b_i$ counts how many have been received from worker $i$. We also initialize $S = \emptyset$ to track which entries have at least one gradient.

**Phase 1 — Gradient Collection.** In this phase, the server receives stochastic gradients from workers and stores them in the table $\{(G_i, b_i)\}_{i=1}^n$. Let $g_j^k$ be the gradient sent by worker $j$ at iteration $k$, computed at $x^{k-\delta_j^k}$ with $\xi_j \sim D_j$. The delays $\delta_j^k$ are not needed to run the algorithm—they will only appear later in the analysis and can be inferred as the gap between the current iterate and the one used to compute $g_j^k$.

Upon receiving $g_j^k$ from worker $j$ (Line 7), the server updates the corresponding table entry and the stochastic gradient counter as follows (Line 8)

$$G_j = G_j + g_j^k \,, \quad b_j = b_j + 1 \,, \quad S = S \cup \{j\} \,.$$

This process continues until $S = [n]$, i.e., the server has collected at least one gradient from every worker. No model updates are performed during this phase, and workers do not receive new points; hence, all stochastic gradients from a given worker are computed at the same point.

**Phase 2 — Sequential Updates.** In this phase, the server performs exactly one model update for each worker $i$, for a total of $n$ updates. Phase 2 starts with the last worker that completed Phase 1, i.e., the worker whose gradient made the table complete. The server first computes an update by averaging the accumulated stochastic gradients in the table $\{(G_i, b_i)\}_{i=1}^n$ and taking a descent step with this estimate (Line 11). The updated model is then sent to this worker (Line 12), the worker is removed from the set $S$, and the iteration counter is incremented (Line 13).

Next, the server must update the remaining $n-1$ workers. These updates are performed sequentially as soon as each worker finishes its current computation. During this waiting period, new gradients may arrive from workers not in $S$—e.g. for example, the last updated worker may send another stochastic gradient before the other workers complete their computation. Since discarding these gradients would waste information, they are instead buffered for the next round.

**Temporary Table Management.** To achieve this, the server maintains a temporary table $\{(G_i^+, b_i^+)\}_{i=1}^n$, initialized to zero (Line 14), together with a set $S^+$ that records which workers have contributed to the table. Whenever a gradient arrives from a worker not in $S$, it is stored in the temporary table (Line 23).

If instead the gradient comes from $j \in S$, the server updates the main table $\{(G_i, b_i)\}_{i=1}^n$ (Line 18), performs a model update using its averages (Line 19), broadcasts the new model to worker $j$ (Line 20), increments the iteration counter, and removes $j$ from $S$ (Line 21).

**Preparing for the Next Round.** Once all workers in $S$ have been updated and Phase 2 ends ($S = \emptyset$), the server copies the temporary table $\{(G_i^+, b_i^+)\}_{i=1}^n$ into the main table $\{(G_i, b_i)\}_{i=1}^n$ (Line 26) and resets $S = S^+$, since these workers already contributed gradients at their updated models. Entries for workers not in $S^+$ remain zero, as the temporary table was initialized with zeros at the start of Phase 2 (Line 14). The server then returns to Phase 1 to begin the next gradient collection round.

The two-phase structure of Ringleader ASGD prevents the unbounded delays of standard asynchronous methods. The precise bound on delays is given in the following lemma

**Lemma 4.1** (Bounded Delay; Proof in Appendix E.2). *In* Ringleader ASGD *(Algorithm 1), the delays $\delta_i^k$ satisfy $\delta_i^k \leq 2n - 2$ , for any worker $i \in [n]$ and any iteration $k \geq 0$.*

## 4.2 Comparison to IA$^2$SGD and Malenia SGD

**IA$^2$SGD.** Our method is a delay-controlled version of IA$^2$SGD. We can recover IA$^2$SGD by removing Phase 1 and Phase 2, and thus perform updates naively—immediately upon gradient arrival. In contrast, our algorithm operates in structured rounds, performing exactly one update per worker in each round, which provides the crucial delay control that IA$^2$SGD lacks.

In IA$^2$SGD, delays for slow workers can grow unboundedly because the server continuously updates the model using gradients from fast workers, causing slow workers to fall increasingly behind. Our method prevents this issue by buffering gradients from fast workers rather than immediately applying these gradients, ensuring that all workers receive updated models within $n$ subsequent iterations.

**Malenia SGD.** Malenia SGD also operates in two phases. In Phase 1, it collects gradients much like our approach, but with a termination condition (4) that requires knowing the noise parameter $\sigma$ and target stationarity $\varepsilon$, making it impractical. In Phase 2, it performs a *synchronous* update by averaging all collected gradients and broadcasting the new model to *all* workers, discarding ongoing computations in the process. In contrast, our method performs Phase 2 *asynchronously*, updating workers sequentially as they finish so that no work is wasted.

Regarding Malenia SGD's termination condition (4), in Appendix I we demonstrate that this condition can be replaced with our simpler requirement of obtaining at least one gradient from every worker. With this modification, Malenia SGD remains optimal in the fixed-time regime (2) while becoming parameter-free, which eliminates the need for prior knowledge of $\sigma$ and $\varepsilon$. Under this parameter-free variant, the only difference between Malenia SGD and Ringleader ASGD lies in Phase 2: we perform updates asynchronously without discarding gradients, while Malenia SGD operates synchronously.

## 5 THEORETICAL RESULTS

Before presenting the theoretical results, we first write the algorithm in a compact form. The gradients for each worker in the table are all computed at the same point; for worker $i$ at iteration $k$, the point is $x^{k-\delta_i^k}$. The update rule can be written compactly as

$$x^{k+1} = x^k - \gamma \bar{g}^k \,,$$

where the gradient estimator $\bar{g}^k$ is defined by

$$\bar{g}^k := \frac{1}{n} \sum_{i=1}^{n} \bar{g}_i^k := \frac{1}{n} \sum_{i=1}^{n} \frac{1}{b_i^k} \sum_{j=1}^{b_i^k} g_i^{k,j} \,.$$

Since multiple gradients may be received from the same worker, we denote by $g_i^{k,j}$ the $j$-th gradient from worker $i$ at iteration $k$. Here the index $j$ corresponds to the i.i.d. sampled data point, and more concretely

$$g_i^{k,j} := \nabla f_i \left( x^{k-\delta_i^k}; \xi_i^{k-\delta_i^k, j} \right) \,.$$

The quantity $b_i^k$ denotes the number of gradients from worker $i$ stored in the table at iteration $k$, i.e., the value of $b_i$ in Lines 11 and 19. Thus, $(b_i^k, \delta_i^k)$ fully determine the method's behavior.

Note that the sequence $\{b_i^k\}$ depends only on the computation times $\{\tau_i\}$ and the algorithm design (i.e., the stopping rule for collecting gradients). Once these are fixed, all $b_i^k$ for every $i \in [n]$ and iteration $k$ are determined. Crucially, the values of $b_i^k$ do not depend on the method's hyperparameters $\gamma$, $x^0$, or on the variance parameter $\sigma$ or the stationarity level $\varepsilon$.

**Iteration Complexity.** We begin by establishing notation for the harmonic means of the batch sizes across rounds

$$B^k = \left( \frac{1}{n} \sum_{i=1}^{n} \frac{1}{b_i^k} \right)^{-1} \,, \quad \text{and} \quad B = \inf_{k \geq 0} B^k.$$

Note that $B \geq 1$, since by the algorithm's design each $b_i^k \geq 1$. A sharper bound on $B$ will be established later in Lemma 5.2. We obtain the iteration complexity from this theorem.

**Theorem 5.1** (Iteration Complexity; Proof in Appendix F). *Under Assumptions 1, 2, and 3, let the stepsize in Ringleader ASGD (Algorithm 1) be*

$$\gamma = \min \left\{ \frac{1}{8nL}, \frac{\varepsilon B}{10L\sigma^2} \right\}.$$

*Then,*

$$\frac{1}{K} \sum_{k=0}^{K-1} \mathbb{E}\left[ \left\| \nabla f(x^k) \right\|^2 \right] \leq \varepsilon \,,$$

*for*

$$K \geq \frac{32nL\Delta}{\varepsilon} + \frac{40L\Delta\sigma^2}{B\varepsilon^2} = \mathcal{O}\left( \frac{nL\Delta}{\varepsilon} \left( 1 + \frac{\sigma^2}{Bn\varepsilon} \right) \right).$$

For parallel and asynchronous methods, iteration complexity is less important than time complexity. What truly matters is how quickly we can finish training. We are willing to perform more iterations and extra computation if it means completing the process faster. Having established the iteration complexity, we now turn to the time complexity.

**Time Complexity.** Since the algorithm operates in rounds with $n$ steps per round, and its iteration complexity is already known, it remains to determine the duration of each round. We have the following lemma

**Lemma 5.2** (Proof in Appendix G.1). *Each block of $n$ consecutive iterations (each round) of Algorithm 1 takes at most $2\tau_n$ seconds. Moreover, we have*

$$B \geq \frac{\tau_n}{2} \left( \frac{1}{n} \sum_{i=1}^{n} \tau_i \right)^{-1} = \frac{\tau_n}{2\tau_{\text{avg}}} \ .$$

Based on this lemma, we derive the time complexity guarantee of our algorithm

**Theorem 5.3** (Proof in Appendix G). *Let the assumptions and parameter choices of Theorem 5.1 hold. Then, under the* fixed computation model (2), Ringleader ASGD *achieves the optimal time complexity*

$$\mathcal{O} \left( \frac{L\Delta}{\varepsilon} \left( \tau_n + \tau_{\text{avg}} \frac{\sigma^2}{n\varepsilon} \right) \right) .$$

The obtained time complexity consists of two key terms that illuminate the algorithm's behavior. The first term depends on the slowest device, which is fundamental since all devices must contribute to solving the problem. The second term, however, involves $\tau_{\text{avg}}$ rather than $\tau_n$ as in Naive Minibatch SGD (see Table 1)—this substitution captures the core benefit of asynchronous execution. Specifically, this advantage becomes pronounced when $\sigma$ is relatively large. Intuitively, in high-noise regimes, collecting many gradients from workers is essential for convergence, and asynchronous methods can leverage faster workers more effectively. Conversely, in low-noise settings, fewer gradient evaluations suffice for good performance, making Naive Minibatch SGD already quite effective and rendering the additional complexity of asynchrony unnecessary.

**Remark 5.4.** *The optimality claim for Theorem 5.3 holds when the smoothness-type constant $L$ in Assumption 2 is within a constant factor of the smoothness $L_f$ used to derive the lower bound (Tyurin & Richtárik, 2024) (Table 1).*

Under this condition, the resulting time complexity matches the lower bound of Tyurin & Richtárik (2024), making Ringleader ASGD the first asynchronous algorithm to achieve optimality under heterogeneous data.

## 6 CONCLUSION

We have introduced Ringleader ASGD, the first asynchronous stochastic gradient method to achieve optimal time complexity under arbitrary data heterogeneity and arbitrarily heterogeneous computation times in distributed learning, without requiring similarity assumptions between workers' datasets.

Its core innovation is a two-phase structure within each round: the model is updated once per worker (for a total of $n$ updates), while a buffering mechanism manages gradient delays and preserves the efficiency of asynchronous execution. By maintaining a gradient table and alternating between gradient collection and sequential updates, Ringleader ASGD prevents the unbounded delays common in naive asynchronous methods. Every gradient received by the server is either used in the current round or stored for future use, ensuring no computation is wasted.

Our analysis shows that Ringleader ASGD matches the optimal time complexity bounds established by Tyurin & Richtárik (2024). In contrast to the optimal but synchronous Malenia SGD method, Ringleader ASGD is asynchronous and requires no prior knowledge of problem parameters in the algorithm design, making it practical for real-world deployments.

Finally, with a minor modification, Ringleader ASGD also achieves optimality in the more general setting of arbitrarily varying computation times (Appendix D).

ACKNOWLEDGMENTS

The research reported in this publication was supported by funding from King Abdullah University of Science and Technology (KAUST): i) KAUST Baseline Research Scheme, ii) CRG Grant ORFS-CRG12-2024-6460, and iii) Center of Excellence for Generative AI, under award number 5940.

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

## A    EXPERIMENTS

To validate our theoretical results we perform a toy simulation.

We consider image classification on MNIST (LeCun et al., 2010) and on Fashion-MNIST (Xiao et al., 2017) with standard normalization for both datasets. To enable equal client sizes, we first trim the dataset so that the total number of examples is divisible by the number of clients $n = 100$. To obtain heterogeneous datasets across clients, we then partition the trimmed set using an *equal-size Dirichlet* procedure with concentration parameter $\alpha = 0.1$ (Yurochkin et al., 2019). For each client $j \in [n]$, we draw proportions $p_j \sim \mathrm{Dirichlet}(\alpha, \dots, \alpha)$ over the classes and compute a rounded class-allocation vector whose entries sum exactly to $N/n$, where $N$ is the trimmed dataset size. This creates non-IID data where each client has a skewed distribution over classes (with $\alpha = 0.1$, clients typically observe only 1-2 classes frequently).

When assigning samples, we take the requested number from each class pool for client $j$. If a class pool does not have enough remaining examples to meet the requested amount, the client receives whatever is left from that class and the shortfall is *topped up* using samples from other classes that still have available examples.

Our model is a two-layer MLP $\mathrm{Linear}(d, 128) \to \mathrm{ReLU} \to \mathrm{Linear}(128, 10)$ trained with mean cross-entropy. Stochastic gradients at the clients use a minibatch of size 4, while reported gradient norms are computed on the *full* dataset.

We emulate heterogeneous compute by assigning each client $i$ a base delay and a random jitter:

$$\tau_i \;=\; i \;+\; |\eta_i|\,, \qquad \eta_i \sim \mathcal{N}(0, i)\,, \quad \text{for all } i \in [n]\,.$$

For each method we tune the stepsize $\gamma$ within a fixed wall-clock budget. We sweep

$$\gamma \in \{0.001,\, 0.005,\, 0.01,\, 0.02,\, 0.05,\, 0.1,\, 0.2,\, 0.5,\, 1.0,\, 2.0\}\,,$$

and then fix the best $\gamma$ per method for evaluation.

We report the full-batch gradient-norm squared $\|\nabla f(x^k)\|^2$, versus wall-clock time. Each method is run 30 times over different seeds. We report the *median* with *interquartile range (IQR)*. To reduce high-frequency noise, we apply a centered moving-average smoothing to the aggregated curves (post-aggregation), while keeping the initial point unchanged.

Figure 1 shows the results. We observe that Ringleader ASGD converges faster compared to Malenia SGD and IA$^2$SGD. Although theory suggests that Ringleader ASGD and Malenia SGD have the same time complexity, in practice Ringleader ASGD benefits from the $n$ updates performed in Phase 2 instead of one synchronous update. This design enables more optimization steps within the same wall-clock budget, which is especially advantageous when updates are sparse.

## B    RELATED WORK

Research on asynchronous stochastic gradient methods dates back to the seminal work of Tsitsiklis et al. (1986), and gained renewed momentum with the introduction of the HOGWILD! algorithm (Recht et al., 2011). HOGWILD! is fundamentally an asynchronous coordinate descent method: updates are performed lock-free with inconsistent reads and writes, and its convergence guarantees rely on sparsity assumptions that are rarely realistic in modern large-scale machine learning. Subsequent refinements of this paradigm include works by J Reddi et al. (2015); Zhao & Li (2016); Mania et al. (2017); Leblond et al. (2018); Nguyen et al. (2018); Zhou et al. (2018), but these works remain tied to the coordinate descent setting with inconsistent memory accesses, and thus differ substantially from our focus.

Closer to our setting are works where updates are based on gradients that are applied consistently. Early contributions, typically under the homogeneous-data assumption (all workers sample from the same distribution), include the work of Agarwal & Duchi (2011), who studied convex objectives, as well as later extensions to the non-convex case such as the work of Lian et al. (2015) and Dutta et al. (2018), the latter analyzing exponentially distributed computation times. Other relevant results in this line include those of Feyzmahdavian et al. (2016); Zheng et al. (2017); Arjevani et al. (2020); Feyzmahdavian & Johansson (2023), all of which assume fixed delays. More recently,

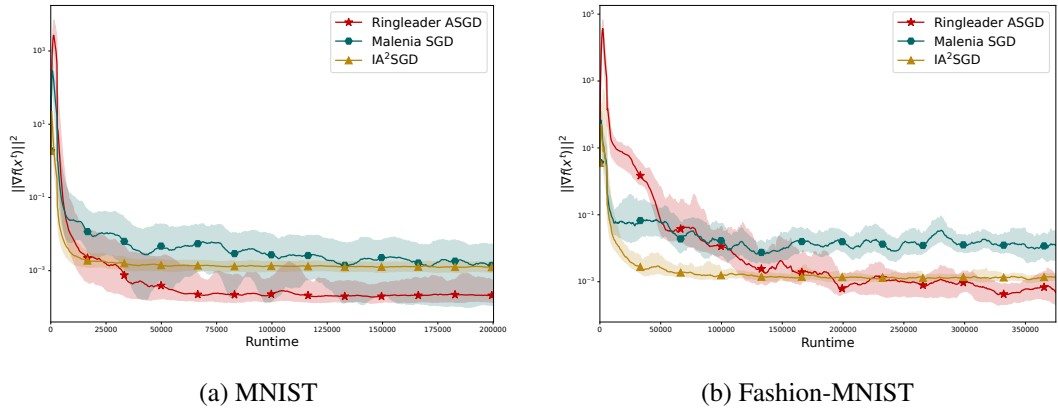

(a) MNIST              (b) Fashion-MNIST

Figure 1: Convergence comparison showing median gradient norm squared $\|\nabla f(x^k)\|^2$ (solid lines) with interquartile ranges (shaded regions) versus wall-clock time, averaged over 30 random seeds. **Setup:** Two-layer MLP with architecture $\mathrm{Linear}(d, 128) \to \mathrm{ReLU} \to \mathrm{Linear}(128, 10)$ trained on (a) MNIST and (b) Fashion-MNIST datasets. **Client delays:** Heterogeneous delays simulated as $\tau_i = i + |\eta_i|$ where $\eta_i \sim \mathcal{N}(0, i)$ for client $i \in [n]$, where we choose $n = 100$. **Results:** With optimally tuned stepsizes, Ringleader ASGD achieves faster convergence than both Malenia SGD and IA$^2$SGD, despite Ringleader ASGD and Malenia SGD having equivalent time complexity guarantees.

delay-adaptive methods have been proposed, aiming to improve performance by down-weighting very stale gradients (Cohen et al., 2021; Koloskova et al., 2022; Mishchenko et al., 2022).

Particularly relevant to our work are asynchronous variants of SAGA. Leblond et al. (2018) developed a shared-parameter version in the spirit of HOGWILD!, while Glasgow & Wootters (2022) studied a distributed setting that employs full gradients, in contrast to our stochastic-gradient perspective.

A large body of recent work investigates asynchronous methods in federated learning (FL), where clients hold data from heterogeneous distributions. Notable contributions include works by Xie et al. (2019); Mishchenko et al. (2022); Koloskova et al. (2022); Wang et al. (2022a;b); Glasgow & Wootters (2022); Fraboni et al. (2023); Zhang et al. (2023); Wang et al. (2024); Islamov et al. (2024); Alahyane et al. (2025).

More broadly, Assran et al. (2020) provide a comprehensive survey of asynchronous optimization methods.

There is another line of work that began with Tyurin & Richtárik (2024), who established lower bounds for parallel methods and proposed optimal synchronous algorithms together with an asynchronous counterpart. Several follow-up papers extended this semi-asynchronous framework to other settings (Tyurin et al., 2024a;b; Tyurin & Richtárik, 2024; Maranjyan et al., 2025a).

Finally, beyond asynchronous approaches, several synchronous methods address system heterogeneity by adapting local training to worker speeds. The canonical method, FedAvg (McMahan et al., 2017), performs multiple local steps on each worker. Variants have adapted the number of local steps to match workers' computation speeds (Li et al., 2020; Maranjyan et al., 2025c), effectively balancing task assignments across heterogeneous systems. More recently, Maranjyan et al. (2025b) proposed adapting the number of local steps dynamically, without requiring prior knowledge of worker speeds.

## C  THE COMPUTATION-ONLY MODEL AND THE ROLE OF ASYNCHRONY

This section clarifies the scope of our modeling assumptions and the specific phenomenon our results target. Asynchrony is designed to eliminate *waiting time*: the idle time that arises when workers have heterogeneous computation speeds or experience straggling behavior due to hardware stalls, load imbalance, or even network delays. A key point is that the goal of asynchrony is *not* to reduce

communication cost, but to ensure that slow or delayed workers do not force faster workers to remain idle.

Critically, even in the *simplest* setting—homogeneous data and *no* communication cost—it was unknown until very recently whether an asynchronous SGD method could match the optimal synchronous rate. In fact, existing asynchronous methods were shown to be *worse* than the optimal synchronous algorithm in this basic regime (Tyurin & Richtárik, 2024). The recent work of Maranjyan et al. (2025d) resolved this foundational case for the first time by showing that, under homogeneous data, an asynchronous method can achieve the same optimal time complexity as the synchronous method Rennala SGD (Tyurin & Richtárik, 2024). Our work extends this understanding to the significantly more challenging heterogeneous-data setting and shows that asynchrony *can* solve the problem it is designed for—and that it can do so *optimally*.

**Asynchrony and stragglers.** Stragglers may be caused by slow computation, device variability, or even *communication delays*. From the server's perspective, a worker whose gradient arrives late because it is still communicating appears identical to one that is slow at computing. Asynchrony ensures that such delays—regardless of source—do not block progress: fast workers continue contributing updates while slow or delayed workers catch up. This ability to eliminate idle time is precisely the purpose of asynchronous methods.

**Asynchrony does *not* reduce communication cost.** Although asynchrony prevents waiting during communication-induced delays, it does *not* reduce the communication overhead itself. Reducing communication cost requires *orthogonal techniques* such as gradient compression, sparsification, quantization, or local-update schemes (e.g., FedAvg (McMahan et al., 2016)). For example, Tyurin et al. (2024b) explicitly study communication-aware training and use compression-based methods to reduce communication time—demonstrating that communication efficiency is a *separate algorithmic axis* that must be combined with, rather than replaced by, asynchrony. Thus, asynchrony resolves the *waiting problem* caused by delays, but not the *communication-cost problem*; addressing the latter requires additional mechanisms.

**Why communication is not modeled explicitly here.** For these reasons, we adopt the standard computation-only model used in essentially all theoretical works on asynchronous SGD (Mishchenko et al., 2022; Koloskova et al., 2022; Tyurin & Richtárik, 2024; Tyurin & Richtárik, 2024; Maranjyan et al., 2025d), which is also the model under which the lower bounds of Tyurin & Richtárik (2024) are derived. Our claims of *optimal time complexity* therefore refer to this shared and well-established model. Studying asynchrony under explicit communication cost—where it must interact with compression, local updates, or buffering—requires new lower bounds and a different theoretical framework, and is beyond the scope of this work.

## D  ARBITRARILY CHANGING COMPUTATION TIMES

In practice, the *fixed computation model* (2) is often not satisfied. The compute power of devices can vary over time due to temporary disconnections, hardware or network delays, fluctuations in processing capacity, or other transient effects (Maranjyan et al., 2025a).

In this section we extend our theory to the more general setting of arbitrarily varying computation times.

### D.1  UNIVERSAL COMPUTATION MODEL

To formalize this setting, we adopt the *universal computation model* introduced by Tyurin (2024).

For each worker $i \in [n]$, we define a *compute power* function
$$p_i : \mathbb{R}_+ \to \mathbb{R}_+ ,$$
assumed nonnegative and continuous almost everywhere (countably many jumps allowed). For any $T_2 \geq T_1 \geq 0$, the number of stochastic gradients *completed* by worker $i$ on $[T_1, T_2]$ is

$$\#\text{gradients in } [T_1, T_2] \;=\; \left\lfloor \int_{T_1}^{T_2} p_i(t)\, dt \right\rfloor .$$

Here, $p_i(t)$ models the worker's time-varying computational ability: smaller values over an interval yield fewer completed gradients, and larger values yield more.

For instance, if worker $i$ remains idle for the first $T$ seconds and then becomes active, this corresponds to $p_i(t) = 0$ for $t \leq T$ and $p_i(t) > 0$ for $t > T$. More generally, $p_i(t)$ may follow periodic or irregular patterns, leading to bursts of activity, pauses, or chaotic changes in compute power. The process $p_i(t)$ may even be random, and all results hold conditional on the realized sample paths of $\{p_i\}$.

The *universal computation model* reduces to the *fixed computation model* (2) when $p_i(t) = 1/\tau_i$ for all $t \geq 0$ and $i \in [n]$. In this case,

$$\#\text{gradients in } [T_1, T_2] = \left\lfloor \frac{T_2 - T_1}{\tau_i} \right\rfloor,$$

meaning that worker $i$ computes one stochastic gradient after $T_1 + \tau_i$ seconds, two gradients after $T_1 + 2\tau_i$ seconds, and so on.

### D.2 TOWARD AN OPTIMAL METHOD

In the general setting of arbitrarily varying computation times, Algorithm 1 is not optimal. To see why, consider the following adversarial timing pattern.

Suppose there are two workers. During one gradient computation by the slower worker, the faster worker computes $s$ gradients. Immediately afterwards, they switch roles: the previously fast worker slows down by a factor of $s$, while the previously slow one speeds up by the same factor. This pattern repeats each time the slower worker finishes a gradient computation.

In this setting, if we run Algorithm 1, the server waits in each Phase 1 for a single gradient from every worker. Thus, the slower worker always contributes only one gradient, and the harmonic mean of the batch sizes satisfies

$$1 \leq B^k \leq 2.$$

From Theorem 5.1, the iteration complexity is

$$\mathcal{O}\left(\frac{nL\Delta}{\varepsilon}\left(1 + \frac{\sigma^2}{Bn\varepsilon}\right)\right).$$

When $\sigma^2/n\varepsilon$ is much larger than $B$, this dependence can be highly suboptimal.

Instead, suppose the server waits until one full round of the above process completes, collecting $s + 1$ gradients from each worker. Then the harmonic mean satisfies $B^k \geq s + 1$, which can be arbitrarily larger than 2. Since in practice both $s$ and $\sigma^2/n\varepsilon$ can be very large, the naive strategy of waiting for only one gradient per worker (as in Algorithm 1) cannot be optimal in the arbitrary-time setting.

### D.3 AN OPTIMAL METHOD

The solution is simple and follows directly from the iteration complexity bound. From

$$\mathcal{O}\left(\frac{nL\Delta}{\varepsilon}\left(1 + \frac{\sigma^2}{Bn\varepsilon}\right)\right),$$

we see that to balance the terms it suffices to ensure

$$B \geq \frac{\sigma^2}{n\varepsilon}.$$

Accordingly, we modify the stopping condition in Phase 1 of Algorithm 1. Instead of requiring the server to receive at least one gradient from each worker, we require the stronger condition used in Malenia SGD, namely

$$\left(\frac{1}{n}\sum_{i=1}^{n}\frac{1}{b_i}\right)^{-1} \geq \max\left\{1, \frac{\sigma^2}{n\varepsilon}\right\}, \tag{4}$$

where $b_i$ is the number of gradients received from worker $i$.

In the low-noise regime, where $\sigma^2/n\varepsilon \leq 1$, the condition reduces to requiring $b_i \geq 1$ for all $i$, so the algorithm coincides with the original Algorithm 1. In the high-noise regime, the algorithm collects more gradients in Phase 1, ensuring that $B$ is sufficiently large for optimal convergence.

With this change, Phase 1 of our algorithm matches that of Malenia SGD. The difference lies in Phase 2: our algorithm continues to use the ongoing gradient computations from all workers to perform $n$ updates, while Malenia SGD discards any unfinished gradients, performs a single update, and then proceeds to the next round.

The following theorem establishes the time complexity of our algorithm under the universal computation model.

**Theorem D.1.** *Under Assumptions 1, 2, and 3, let the stepsize in* Ringleader ASGD *be*

$$\gamma = \frac{1}{10nL} \ .$$

*Then, under the* universal computation model, Ringleader ASGD *finds an $\varepsilon$–stationary point within at most $T^K$ seconds, where*

$$K \ := \ \left\lceil \frac{160L\Delta}{\varepsilon} \right\rceil,$$

*and $T^K$ denotes the $K$-th element of the recursively defined sequence*

$$T^k \ = \ \min\left\{ T \geq 0 : \left( \frac{1}{n} \sum_{i=1}^n \left\lfloor \int_{T_{k-1}}^T p_i(t)\,dt \right\rfloor^{-1} \right)^{-1} \geq \max\left\{1, \frac{\sigma^2}{n\varepsilon}\right\} \right\},$$

*for all $k \geq 1$, with initialization $T^0 = 0$.*

This result matches the lower bound derived by Tyurin (2024), and therefore the proposed method is optimal.

*Proof.* Under the condition in (4), each gradient-type step of the algorithm satisfies

$$B^k = \left( \frac{1}{n} \sum_{i=1}^n \frac{1}{b_i^k} \right)^{-1} \geq \max\left\{1, \frac{\sigma^2}{n\varepsilon}\right\}.$$

In Theorem 5.1, instead of using $B$ we can substitute any valid lower bound. Here we choose

$$B = \max\left\{1, \frac{\sigma^2}{n\varepsilon}\right\}.$$

With this substitution, the iteration complexity becomes

$$K = \frac{80\,nL\Delta}{\varepsilon} \ .$$

To derive the time complexity, consider the time required to perform $n$ iterations. Each block of $n$ updates occurs in Phase 2 following the Phase 1 gradient collection. Starting from time $T = 0$, Phase 1 ends once the accumulated number of gradients satisfies condition (4), which occurs at time

$$T_+^1 = \min\left\{ T \geq 0 : \left( \frac{1}{n} \sum_{i=1}^n \left\lfloor \int_0^T p_i(t)\,dt \right\rfloor^{-1} \right)^{-1} \geq \max\left\{1, \frac{\sigma^2}{n\varepsilon}\right\} \right\}.$$

After Phase 1, to complete $n$ updates in Phase 2 we must wait for the ongoing computations to finish. This requires at most

$$T^1 = \min\left\{ T \geq 0 : \left( \frac{1}{n} \sum_{i=1}^n \left\lfloor \int_{T_+^1}^T p_i(t)\,dt \right\rfloor^{-1} \right)^{-1} \geq 1 \right\}.$$

Thus, the total time to complete all $K$ iterations is bounded by

$$T^{\lceil 2K/n \rceil} ,$$

where the sequence $\{T^k\}_{k \geq 0}$ is defined recursively as

$$T^k = \min \left\{ T \geq 0 : \left( \frac{1}{n} \sum_{i=1}^{n} \left\lfloor \int_{T_{k-1}}^{T} p_i(t)\, dt \right\rfloor^{-1} \right)^{-1} \geq \max \left\{ 1, \frac{\sigma^2}{n\varepsilon} \right\} \right\}, \qquad T^0 = 0 .$$

$\square$

## E AUXILIARY LEMMAS

Here we provide proofs of lemmas omitted from the main text, along with auxiliary results that will be used later.

### E.1 RELATIONS BETWEEN SMOOTHNESS CONSTANTS

We begin with a lemma relating the different smoothness constants.

**Lemma E.1** (Smoothness Bounds). *Suppose Assumption 2 holds with constant $L > 0$. Then $f$ is $L_f$–smooth with $L_f \leq L$. Moreover, if each $f_i$ is $L_{f_i}$–smooth, then Assumption 2 is satisfied, and we have*

$$L_f \;\leq\; L \;\leq\; \sqrt{\frac{1}{n} \sum_{i=1}^{n} L_{f_i}^2} \;\leq\; \max_{i \in [n]} L_{f_i} =: L_{\max} .$$

*Finally, if all $f_i$ are identical, i.e., $f_i = f$ for all $i \in [n]$, then $L = L_f$.*

Recall from Assumption 2 that we assumed the following generalized smoothness condition: for some constant $L > 0$ and for all $x \in \mathbb{R}^d$ and $y_1, \ldots, y_n \in \mathbb{R}^d$,

$$\left\| \nabla f(x) - \frac{1}{n} \sum_{i=1}^{n} \nabla f_i(y_i) \right\|^2 \;\leq\; \frac{L^2}{n} \sum_{i=1}^{n} \|x - y_i\|^2 . \tag{5}$$

Recall that a function $\phi$ is called $L_\phi$–smooth if

$$\|\nabla \phi(x) - \nabla \phi(y)\| \leq L_\phi \|x - y\|, \quad \forall x, y \in \mathbb{R}^d .$$

Here $L_\phi$ denotes the minimal such constant. We are ready to prove the lemma.

*Proof.* For the first inequality, take $y_1 = \cdots = y_n = y$. Then (5) reduces to

$$\|\nabla f(x) - \nabla f(y)\|^2 \leq L^2 \|x - y\|^2 ,$$

so $f$ is $L$–smooth. By definition of $L_f$ as the minimal smoothness constant, this implies $L_f \leq L$.

For the second inequality, by the triangle inequality, then by the smoothness of each $f_i$, and finally by Cauchy–Schwarz,

$$\left\| \nabla f(x) - \frac{1}{n} \sum_{i=1}^{n} \nabla f_i(y_i) \right\| \leq \frac{1}{n} \sum_{i=1}^{n} \|\nabla f_i(x) - \nabla f_i(y_i)\| \leq \frac{1}{n} \sum_{i=1}^{n} L_{f_i} \|x - y_i\|$$

$$\leq \sqrt{\frac{1}{n} \sum_{i=1}^{n} L_{f_i}^2} \; \sqrt{\frac{1}{n} \sum_{i=1}^{n} \|x - y_i\|^2} .$$

Squaring both sides shows that (5) holds with $L = \sqrt{\frac{1}{n} \sum_{i=1}^{n} L_{f_i}^2}$ .

Finally, suppose all $f_i$ are identical: $f_i \equiv f$ for all $i$. Then

$$\left\| \nabla f(x) - \frac{1}{n} \sum_{i=1}^{n} \nabla f(y_i) \right\| \leq \frac{1}{n} \sum_{i=1}^{n} \| \nabla f(x) - \nabla f(y_i) \| \leq \frac{L_f}{n} \sum_{i=1}^{n} \| x - y_i \|$$

$$\leq L_f \sqrt{\frac{1}{n} \sum_{i=1}^{n} \| x - y_i \|^2} ,$$

where the last step uses Cauchy–Schwarz. Squaring both sides yields

$$\left\| \nabla f(x) - \frac{1}{n} \sum_{i=1}^{n} \nabla f(y_i) \right\|^2 \leq \frac{L_f^2}{n} \sum_{i=1}^{n} \| x - y_i \|^2 ,$$

i.e., (5) holds with $L \leq L_f$. Combined with $L_f \leq L$, we conclude $L = L_f$. $\qquad\square$

### E.2 PROOF OF LEMMA 4.1

The following lemma gives a bound on the delays of Ringleader ASGD.

**Lemma 4.1** (Bounded Delay). In Ringleader ASGD (Algorithm 1), the delays $\delta_i^k$ satisfy

$$\delta_i^k \leq 2n - 2 ,$$

for any worker $i \in [n]$ and any iteration $k \geq 0$.

*Proof.* We prove this by analyzing the structure of Ringleader ASGD. The algorithm operates in rounds, where each round consists of Phase 1 (gradient collection) followed by Phase 2 (sequential updates). In each Phase 2, the server performs exactly $n$ updates, one for each worker. Phase 2 begins at iterations $0, n, 2n, 3n, \ldots$, i.e., at multiples of $n$.

**First round (iterations $0$ to $n - 1$):** Initially, all workers compute gradients at point $x^0$, so during iterations $0, 1, \ldots, n - 1$, the server receives gradients computed at $x^0$. For any iteration $k$ in this range, the server processes gradients $\nabla f_i(x^{k - \delta_i^k})$, so $\delta_i^k = k \leq n - 1$. Thus, delays simply increment during Phase 2.

At the end of this round, each worker $i$ has received a new model $x^j$ for some $j \in \{1, 2, \ldots, n\}$, and these update iterations are distinct across workers.

**Second round (iterations $n$ to $2n - 1$):** At the start of the second Phase 2 (at iteration $n$), the gradient table contains gradients $\nabla f_i(x^{n - \delta_i^n})$ where $\delta_i^n \in \{0, 1, \ldots, n - 1\}$. These delay values are distinct across workers since each worker received its update at a different iteration in the previous round.

During iterations $n$ to $2n - 1$, these delays increase by $1$ at each iteration for the same reason as in the first Phase 2, giving $\delta_i^{2n-1} \in \{n - 1, n, \ldots, 2n - 2\}$ at the end of this round. At the same time, all workers receive new points to compute gradients from, so during the next Phase 2, the delays will again be distinct for all workers and in $\{0, 1, \ldots, n - 1\}$.

**General pattern:** By induction, at the beginning of each round starting at iteration $cn$ (for integer $c \geq 1$), the delays $\delta_i^{cn}$ take distinct values in $\{0, 1, \ldots, n - 1\}$. During each Phase 2, these delays increase by at most $n - 1$, giving the bound:

$$\delta_i^k \leq (n - 1) + (n - 1) = 2n - 2.$$

$\qquad\square$

### E.3 VARIANCE TERM

The following lemma bounds the variance of the gradient estimator in Ringleader ASGD.

**Lemma E.2** (Variance Bound). *Under Assumption 1, the following variance-type inequality holds for the gradient estimator used in Algorithm 1:*

$$\mathbb{E}\left[\left\|\bar{g}^k - \frac{1}{n}\sum_{i=1}^n \nabla f_i\left(x^{k-\delta_i^k}\right)\right\|^2\right] \leq \frac{\sigma^2}{B^k n} \ .$$

*Proof.* Recall that the gradient estimator is defined as

$$\bar{g}^k = \frac{1}{n}\sum_{i=1}^n \bar{g}_i^k = \frac{1}{n}\sum_{i=1}^n \frac{1}{b_i^k}\sum_{j=1}^{b_i^k} g_i^{k,j} = \frac{1}{n}\sum_{i=1}^n \frac{1}{b_i^k}\sum_{j=1}^{b_i^k} \nabla f_i\left(x^{k-\delta_i^k}; \xi_i^{k-\delta_i^k,j}\right) \ .$$

Let $\mathcal{F}^k$ denote the sigma-field containing all randomness up to the start of the current round, i.e., up to iteration $k - (k \bmod n)$. Conditioning on $\mathcal{F}^k$, the evaluation points $x^{k-\delta_i^k}$ are deterministic, and the stochastic gradients $g_i^{k,j}$ are independent across both workers $i$ and samples $j$.

Using the law of total expectation and the independence of stochastic gradients, we have

$$\mathbb{E}\left[\left\|\bar{g}^k - \frac{1}{n}\sum_{i=1}^n \nabla f_i\left(x^{k-\delta_i^k}\right)\right\|^2\right] = \mathbb{E}\left[\mathbb{E}\left[\left\|\bar{g}^k - \frac{1}{n}\sum_{i=1}^n \nabla f_i\left(x^{k-\delta_i^k}\right)\right\|^2 \ \middle| \ \mathcal{F}^k\right]\right]$$

$$= \mathbb{E}\left[\frac{1}{n^2}\sum_{i=1}^n \mathbb{E}\left[\left\|\bar{g}_i^k - \nabla f_i\left(x^{k-\delta_i^k}\right)\right\|^2 \ \middle| \ \mathcal{F}^k\right]\right] .$$

For each worker $i$, the conditional variance of the minibatch gradient estimator is

$$\mathbb{E}\left[\left\|\bar{g}_i^k - \nabla f_i\left(x^{k-\delta_i^k}\right)\right\|^2 \ \middle| \ \mathcal{F}^k\right] = \mathbb{E}\left[\left\|\frac{1}{b_i^k}\sum_{j=1}^{b_i^k} g_i^{k,j} - \nabla f_i\left(x^{k-\delta_i^k}\right)\right\|^2 \ \middle| \ \mathcal{F}^k\right]$$

$$= \frac{1}{b_i^k}\mathbb{E}\left[\left\|g_i^{k,1} - \nabla f_i\left(x^{k-\delta_i^k}\right)\right\|^2 \ \middle| \ \mathcal{F}^k\right] \leq \frac{\sigma^2}{b_i^k} \ ,$$

where the last inequality follows from Assumption 1.

Combining these results, we get

$$\mathbb{E}\left[\left\|\bar{g}^k - \frac{1}{n}\sum_{i=1}^n \nabla f_i\left(x^{k-\delta_i^k}\right)\right\|^2\right] \leq \frac{1}{n^2}\sum_{i=1}^n \frac{\sigma^2}{b_i^k} = \frac{\sigma^2}{n}\frac{1}{n}\sum_{i=1}^n \frac{1}{b_i^k} = \frac{\sigma^2}{B^k n} \ ,$$

where the last equality uses the definition of the harmonic mean $B^k = \left(\frac{1}{n}\sum_{i=1}^n \frac{1}{b_i^k}\right)^{-1}$. $\qquad\square$

# F    PROOF OF THEOREM 5.1

Our convergence analysis follows the structured approach employed by Maranjyan et al. (2025d), which decomposes the proof into two key components: a descent lemma that tracks the progress of the objective function and a residual estimation lemma that controls the accumulated delays in the system.

The first lemma quantifies how much the objective function decreases at each iteration, accounting for both the standard gradient descent progress and the additional complexities introduced by asynchronous updates.

**Lemma F.1** (Descent Lemma; Proof in Appendix F.1). *Under Assumptions 1 and 2, if the stepsize in Algorithm 1 satisfies $\gamma \leq 1/4L$, then the following inequality holds*

$$\mathbb{E}\left[f\left(x^{k+1}\right)\right] \leq \mathbb{E}\left[f\left(x^k\right)\right] - \frac{\gamma}{2}\mathbb{E}\left[\left\|\nabla f\left(x^k\right)\right\|^2\right] - \frac{\gamma}{4}\mathbb{E}\left[\left\|\frac{1}{n}\sum_{i=1}^n \nabla f_i\left(x^{k-\delta_i^k}\right)\right\|^2\right]$$

$$+ \frac{\gamma L^2}{2n}\sum_{i=1}^n \mathbb{E}\left[\left\|x^k - x^{k-\delta_i^k}\right\|^2\right] + \frac{3\gamma^2 L\sigma^2}{2B}$$

$$+ \gamma^2 L \sum_{\ell=k-(k \bmod n)}^{k-1} \mathbb{E}\left[\left\|\frac{1}{n}\sum_{i=1}^n \nabla f_i\left(x^{\ell-\delta_i^\ell}\right)\right\|^2\right].$$

This descent lemma shares a similar structure with its counterpart in the homogeneous setting analyzed by Maranjyan et al. (2025d), but with a crucial additional term. The final summation term in the upper bound captures the effect of using stale gradients from the gradient table—a phenomenon we refer to as "table delay". This term is absent in the homogeneous case because no gradient table is maintained. Indeed, when $n = 1$, our setting reduces to the homogeneous case, the gradient table becomes unnecessary, and this additional term vanishes, recovering the original descent lemma established by Maranjyan et al. (2025d).

Next, similar to (Maranjyan et al., 2025d), we derive a lemma to bound the term involving the difference between current and old points.

**Lemma F.2** (Residual Estimation; Proof in Appendix F.2). *Under Assumption 1, the iterates of Ringleader ASGD (Algorithm 1) with stepsize $\gamma \leq 1/4nL$ satisfy the following bound*

$$\frac{1}{K}\sum_{k=0}^{K-1}\frac{1}{n}\sum_{i=1}^n \mathbb{E}\left[\left\|x^k - x^{k-\delta_i^k}\right\|^2\right] \leq \frac{2\gamma n}{LK}\sum_{k=0}^{K-1}\mathbb{E}\left[\left\|\frac{1}{n}\sum_{j=1}^n \nabla f_j\left(x^{k-\delta_j^k}\right)\right\|^2\right] + \frac{2\gamma\sigma^2}{LB}.$$

Finally, we get the iteration complexity combining these two lemmas.

**Theorem 5.1** (Iteration Complexity). Under Assumptions 1, 2, and 3, let the stepsize in Ringleader ASGD (Algorithm 1) be

$$\gamma = \min\left\{\frac{1}{8nL}, \frac{\varepsilon B}{10L\sigma^2}\right\}.$$

Then,

$$\frac{1}{K}\sum_{k=0}^{K-1}\mathbb{E}\left[\left\|\nabla f\left(x^k\right)\right\|^2\right] \leq \varepsilon,$$

for

$$K \geq \frac{32nL\Delta}{\varepsilon} + \frac{40L\Delta\sigma^2}{B\varepsilon^2} = \mathcal{O}\left(\frac{nL\Delta}{\varepsilon}\left(1 + \frac{\sigma^2}{Bn\varepsilon}\right)\right).$$

*Proof.* We start by averaging the inequality from Lemma F.1 over $K$ iterations and dividing both sides by $\gamma/2$

$$\frac{1}{K}\sum_{k=0}^{K-1}\mathbb{E}\left[\left\|\nabla f\left(x^k\right)\right\|^2\right] + \frac{1}{2K}\sum_{k=0}^{K-1}\mathbb{E}\left[\left\|\frac{1}{n}\sum_{i=1}^n \nabla f_i\left(x^{k-\delta_i^k}\right)\right\|^2\right]$$

$$\leq \frac{2\Delta}{\gamma K} + \frac{3\gamma L\sigma^2}{B}$$

$$+ \frac{L^2}{n}\frac{1}{K}\sum_{k=0}^{K-1}\sum_{i=1}^n \mathbb{E}\left[\left\|x^k - x^{k-\delta_i^k}\right\|^2\right]$$

$$+ 2\gamma L\frac{1}{K}\sum_{k=0}^{K-1}\sum_{\ell=k-(k \bmod n)}^{k-1} \mathbb{E}\left[\left\|\frac{1}{n}\sum_{i=1}^n \nabla f_i\left(x^{\ell-\delta_i^\ell}\right)\right\|^2\right].$$

We now bound the third term on the right using Lemma F.2

$$\frac{1}{K}\sum_{k=0}^{K-1}\mathbb{E}\left[\left\|\nabla f\left(x^k\right)\right\|^2\right]+\frac{1}{2K}\sum_{k=0}^{K-1}\mathbb{E}\left[\left\|\frac{1}{n}\sum_{i=1}^{n}\nabla f_i\left(x^{k-\delta_i^k}\right)\right\|^2\right]$$

$$\leq\frac{2\Delta}{\gamma K}+\frac{3\gamma L\sigma^2}{B}+\frac{2\gamma L\sigma^2}{B}$$

$$+2\gamma Ln\frac{1}{K}\sum_{k=0}^{K-1}\mathbb{E}\left[\left\|\frac{1}{n}\sum_{j=1}^{n}\nabla f_j\left(x^{k-\delta_j^k}\right)\right\|^2\right]$$

$$+2\gamma L\frac{1}{K}\sum_{k=0}^{K-1}\sum_{\ell=k-(k\bmod n)}^{k-1}\mathbb{E}\left[\left\|\frac{1}{n}\sum_{i=1}^{n}\nabla f_i\left(x^{\ell-\delta_i^\ell}\right)\right\|^2\right]$$

$$\leq\frac{2\Delta}{\gamma K}+\frac{5\gamma L\sigma^2}{B}$$

$$+2\gamma Ln\frac{1}{K}\sum_{k=0}^{K-1}\mathbb{E}\left[\left\|\frac{1}{n}\sum_{j=1}^{n}\nabla f_j\left(x^{k-\delta_j^k}\right)\right\|^2\right]$$

$$+2\gamma Ln\frac{1}{K}\sum_{k=0}^{K-1}\mathbb{E}\left[\left\|\frac{1}{n}\sum_{i=1}^{n}\nabla f_i\left(x^{k-\delta_i^k}\right)\right\|^2\right].$$

Now, using the bound $\gamma\leq 1/8nL$, we obtain

$$\frac{1}{K}\sum_{k=0}^{K-1}\mathbb{E}\left[\left\|\nabla f\left(x^k\right)\right\|^2\right]\leq\frac{2\Delta}{\gamma K}+\frac{5\gamma L\sigma^2}{B}.$$

Finally, plugging in the stepsize and the expression for $K$ ensures the right-hand side is bounded by $\varepsilon$. $\qquad\square$

## F.1 PROOF OF LEMMA F.1

We now prove the descent lemma.

**Lemma F.1** (Descent Lemma). *Under Assumptions 1 and 2, if the stepsize in Algorithm 1 satisfies $\gamma\leq 1/4L$, then the following inequality holds*

$$\mathbb{E}\left[f\left(x^{k+1}\right)\right]\leq\mathbb{E}\left[f\left(x^k\right)\right]-\frac{\gamma}{2}\mathbb{E}\left[\left\|\nabla f\left(x^k\right)\right\|^2\right]-\frac{\gamma}{4}\mathbb{E}\left[\left\|\frac{1}{n}\sum_{i=1}^{n}\nabla f_i\left(x^{k-\delta_i^k}\right)\right\|^2\right]$$

$$+\frac{\gamma L^2}{2n}\sum_{i=1}^{n}\mathbb{E}\left[\left\|x^k-x^{k-\delta_i^k}\right\|^2\right]+\frac{3\gamma^2 L\sigma^2}{2B}$$

$$+\gamma^2 L\sum_{\ell=k-(k\bmod n)}^{k-1}\mathbb{E}\left[\left\|\frac{1}{n}\sum_{i=1}^{n}\nabla f_i\left(x^{\ell-\delta_i^\ell}\right)\right\|^2\right].$$

*Proof.* Some proof techniques are adapted from the works of Maranjyan et al. (2025d) and Wang et al. (2025).

From Assumption 2 and Lemma E.1, we know that $f$ is $L$–smooth. Therefore, the following standard inequality holds (Nesterov, 2018)

$$\mathbb{E}\left[f(x^{k+1})\right]\leq\mathbb{E}\left[f(x^k)-\gamma\left\langle\nabla f(x^k),\bar{g}^k\right\rangle+\frac{L\gamma^2}{2}\left\|\bar{g}^k\right\|^2\right]. \tag{6}$$

Recall that the gradient estimator is defined as

$$\bar{g}^k = \frac{1}{n} \sum_{i=1}^{n} \bar{g}_i^k = \frac{1}{n} \sum_{i=1}^{n} \frac{1}{b_i^k} \sum_{j=1}^{b_i^k} g_i^{k,j} \ .$$

Let $\mathcal{F}^k$ denote the sigma-field containing all randomness up to the start of the current Phase 2, i.e., up to iteration $k - (k \bmod n)$. A key observation is that all gradients in the current gradient table were computed and received during the current round. Since these gradients were computed at points from previous iterations within the current round, we have $k - \delta_i^k \leq k - (k \bmod n)$ for all $i \in [n]$. Conditioning on $\mathcal{F}^k$, the points $x^{k-\delta_i^k}$ are deterministic. Therefore, we can compute the conditional expectation of the gradient estimator:

$$\mathbb{E}\left[\bar{g}^k \mid \mathcal{F}^k\right] = \frac{1}{n} \sum_{i=1}^{n} \frac{1}{b_i^k} \sum_{j=1}^{b_i^k} \mathbb{E}\left[g_i^{k,j} \mid \mathcal{F}^k\right] = \frac{1}{n} \sum_{i=1}^{n} \nabla f_i\left(x^{k-\delta_i^k}\right).$$

The last equality follows from the unbiasedness of the stochastic gradient estimator (Assumption 1).

Using this conditional expectation and the law of total expectation, we can now simplify the inner product term in (6):

$$
\begin{aligned}
\mathbb{E}\left[\langle \nabla f\left(x^k\right), \bar{g}^k\rangle\right] &= \mathbb{E}\left[\left\langle \nabla f\left(x^k\right), \frac{1}{n} \sum_{i=1}^{n} \nabla f_i\left(x^{k-\delta_i^k}\right)\right\rangle\right] \\
&\quad + \mathbb{E}\left[\left\langle \nabla f\left(x^k\right), \bar{g}^k - \frac{1}{n} \sum_{i=1}^{n} \nabla f_i\left(x^{k-\delta_i^k}\right)\right\rangle\right] \\
&= \mathbb{E}\left[\left\langle \nabla f\left(x^k\right), \frac{1}{n} \sum_{i=1}^{n} \nabla f_i\left(x^{k-\delta_i^k}\right)\right\rangle\right] \\
&\quad + \mathbb{E}\left[\left\langle \nabla f\left(x^k\right) - \nabla f\left(x^{k-(k \bmod n)}\right), \bar{g}^k - \frac{1}{n} \sum_{i=1}^{n} \nabla f_i\left(x^{k-\delta_i^k}\right)\right\rangle\right] \\
&\quad + \mathbb{E}\left[\left\langle \nabla f\left(x^{k-(k \bmod n)}\right), \bar{g}^k - \frac{1}{n} \sum_{i=1}^{n} \nabla f_i\left(x^{k-\delta_i^k}\right)\right\rangle\right] \\
&= \underbrace{\mathbb{E}\left[\left\langle \nabla f\left(x^k\right), \frac{1}{n} \sum_{i=1}^{n} \nabla f_i\left(x^{k-\delta_i^k}\right)\right\rangle\right]}_{T_1} \\
&\quad + \underbrace{\mathbb{E}\left[\left\langle \nabla f\left(x^k\right) - \nabla f\left(x^{k-(k \bmod n)}\right), \bar{g}^k - \frac{1}{n} \sum_{i=1}^{n} \nabla f_i\left(x^{k-\delta_i^k}\right)\right\rangle\right]}_{T_2}.
\end{aligned}
$$

Next, using Assumption 2, we have

$$
\begin{aligned}
2T_1 &= \mathbb{E}\left[2\left\langle \nabla f\left(x^k\right), \frac{1}{n} \sum_{i=1}^{n} \nabla f_i\left(x^{k-\delta_i^k}\right)\right\rangle\right] \\
&= \mathbb{E}\left[\left\|\nabla f\left(x^k\right)\right\|^2 + \left\|\frac{1}{n} \sum_{i=1}^{n} \nabla f_i\left(x^{k-\delta_i^k}\right)\right\|^2\right] - \mathbb{E}\left[\left\|\nabla f\left(x^k\right) - \frac{1}{n} \sum_{i=1}^{n} \nabla f_i\left(x^{k-\delta_i^k}\right)\right\|^2\right] \\
&\geq \mathbb{E}\left[\left\|\nabla f\left(x^k\right)\right\|^2\right] + \mathbb{E}\left[\left\|\frac{1}{n} \sum_{i=1}^{n} \nabla f_i\left(x^{k-\delta_i^k}\right)\right\|^2\right] - \frac{L^2}{n} \sum_{i=1}^{n} \mathbb{E}\left[\left\|x^k - x^{k-\delta_i^k}\right\|^2\right].
\end{aligned}
$$

Next, we analyze $T_2$

$$
\begin{aligned}
T_2 &= \mathbb{E}\left[\left\langle \nabla f\left(x^k\right) - \nabla f\left(x^{k-(k \bmod n)}\right), \bar{g}^k - \frac{1}{n}\sum_{i=1}^n \nabla f_i\left(x^{k-\delta_i^k}\right)\right\rangle\right] \\
&\geq -\mathbb{E}\left[\left\|\nabla f\left(x^k\right) - \nabla f\left(x^{k-(k \bmod n)}\right)\right\|\left\|\bar{g}^k - \frac{1}{n}\sum_{i=1}^n \nabla f_i\left(x^{k-\delta_i^k}\right)\right\|\right] \\
&\geq -L\mathbb{E}\left[\left\|x^k - x^{k-(k \bmod n)}\right\|\left\|\bar{g}^k - \frac{1}{n}\sum_{i=1}^n \nabla f_i\left(x^{k-\delta_i^k}\right)\right\|\right] \\
&= -L\mathbb{E}\left[\left\|\gamma\sum_{\ell=k-(k \bmod n)}^{k-1} \bar{g}^\ell\right\|\left\|\bar{g}^k - \frac{1}{n}\sum_{i=1}^n \nabla f_i\left(x^{k-\delta_i^k}\right)\right\|\right] \\
&\geq -L\gamma\sum_{\ell=k-(k \bmod n)}^{k-1} \mathbb{E}\left[\left\|\bar{g}^\ell\right\|\left\|\bar{g}^k - \frac{1}{n}\sum_{i=1}^n \nabla f_i\left(x^{k-\delta_i^k}\right)\right\|\right] \\
&\geq -L\gamma\sum_{\ell=k-(k \bmod n)}^{k-1} \frac{1}{2}\left(\mathbb{E}\left[\left\|\bar{g}^\ell\right\|^2\right] + \mathbb{E}\left[\left\|\bar{g}^k - \frac{1}{n}\sum_{i=1}^n \nabla f_i\left(x^{k-\delta_i^k}\right)\right\|^2\right]\right) \\
&\geq -\frac{L\gamma}{2}\sum_{\ell=k-(k \bmod n)}^{k-1} \mathbb{E}\left[\left\|\bar{g}^\ell\right\|^2\right] - (k \bmod n)\frac{L\gamma\sigma^2}{2B^k n} \\
&\geq -\frac{L\gamma}{2}\sum_{\ell=k-(k \bmod n)}^{k-1} \mathbb{E}\left[\left\|\bar{g}^\ell\right\|^2\right] - \frac{L\gamma\sigma^2}{2B^k} .
\end{aligned}
$$

The inequalities follow from the Cauchy-Schwarz inequality, $L$–smoothness of $f$, the triangle inequality, Young's inequality, Lemma E.2, and finally $(k \bmod n) \leq n-1 < n$.

It remains to bound the term $\mathbb{E}\left[\left\|\bar{g}^k\right\|^2\right]$. Using Young's inequality, we have

$$
\begin{aligned}
\mathbb{E}\left[\left\|\bar{g}^k\right\|^2\right] &\leq 2\mathbb{E}\left[\left\|\frac{1}{n}\sum_{i=1}^n \nabla f_i\left(x^{k-\delta_i^k}\right)\right\|^2\right] + 2\mathbb{E}\left[\left\|\bar{g}^k - \frac{1}{n}\sum_{i=1}^n \nabla f_i\left(x^{k-\delta_i^k}\right)\right\|^2\right] \\
&\leq 2\mathbb{E}\left[\left\|\frac{1}{n}\sum_{i=1}^n \nabla f_i\left(x^{k-\delta_i^k}\right)\right\|^2\right] + \frac{2\sigma^2}{B^k n} ,
\end{aligned}
$$

where in the last step we used Lemma E.2.

Now, by combining all terms in (6), we obtain

$$
\begin{aligned}
\mathbb{E}\left[f\left(x^{k+1}\right)\right] &\leq \mathbb{E}\left[f\left(x^{k}\right)\right] - \frac{\gamma}{2}\mathbb{E}\left[\left\|\nabla f\left(x^{k}\right)\right\|^{2}\right] - \frac{\gamma}{2}\mathbb{E}\left[\left\|\frac{1}{n}\sum_{i=1}^{n}\nabla f_{i}\left(x^{k-\delta_{i}^{k}}\right)\right\|^{2}\right] \\
&\quad + \frac{\gamma L^{2}}{2n}\sum_{i=1}^{n}\mathbb{E}\left[\left\|x^{k}-x^{k-\delta_{i}^{k}}\right\|^{2}\right] \\
&\quad + \frac{\gamma^{2}L}{2}\sum_{\ell=k-(k\bmod n)}^{k-1}\mathbb{E}\left[\left\|\bar{g}^{\ell}\right\|^{2}\right] + \frac{\gamma^{2}L\sigma^{2}}{2B^{k}} + \frac{\gamma^{2}L}{2}\mathbb{E}\left[\left\|\bar{g}^{k}\right\|^{2}\right] \\
&\leq \mathbb{E}\left[f\left(x^{k}\right)\right] - \frac{\gamma}{2}\mathbb{E}\left[\left\|\nabla f\left(x^{k}\right)\right\|^{2}\right] - \frac{\gamma}{2}\mathbb{E}\left[\left\|\frac{1}{n}\sum_{i=1}^{n}\nabla f_{i}\left(x^{k-\delta_{i}^{k}}\right)\right\|^{2}\right] \\
&\quad + \frac{\gamma L^{2}}{2n}\sum_{i=1}^{n}\mathbb{E}\left[\left\|x^{k}-x^{k-\delta_{i}^{k}}\right\|^{2}\right] \\
&\quad + \gamma^{2}L\sum_{\ell=k-(k\bmod n)}^{k-1}\mathbb{E}\left[\left\|\frac{1}{n}\sum_{i=1}^{n}\nabla f_{i}\left(x^{\ell-\delta_{i}^{\ell}}\right)\right\|^{2}\right] \\
&\quad + \gamma^{2}L\mathbb{E}\left[\left\|\frac{1}{n}\sum_{i=1}^{n}\nabla f_{i}\left(x^{k-\delta_{i}^{k}}\right)\right\|^{2}\right] \\
&\quad + \gamma^{2}L\sum_{\ell=k-(k\bmod n)}^{k-1}\frac{\sigma^{2}}{B^{\ell}n} + \frac{\gamma^{2}L\sigma^{2}}{2B^{k}} + \frac{\gamma^{2}L\sigma^{2}}{B^{k}n}\,.
\end{aligned}
$$

This completes the proof under the stepsize condition $\gamma \leq 1/4L$ and $B := \inf_{k\geq 0}B^{k}$. $\qquad\square$

## F.2  PROOF OF LEMMA F.2

The following lemma provides an upper bound on the residual error due to delays.

**Lemma F.2** (Residual Estimation). Under Assumption 1, the iterates of Ringleader ASGD (Algorithm 1) with stepsize $\gamma \leq 1/4nL$ satisfy the following bound

$$
\frac{1}{K}\sum_{k=0}^{K-1}\frac{1}{n}\sum_{i=1}^{n}\mathbb{E}\left[\left\|x^{k}-x^{k-\delta_{i}^{k}}\right\|^{2}\right] \leq \frac{2\gamma n}{LK}\sum_{k=0}^{K-1}\mathbb{E}\left[\left\|\frac{1}{n}\sum_{j=1}^{n}\nabla f_{j}\left(x^{k-\delta_{j}^{k}}\right)\right\|^{2}\right] + \frac{2\gamma\sigma^{2}}{LB}\,.
$$

*Proof.* By Young's inequality, we have

$$\mathbb{E}\left[\left\|x^k - x^{k-\delta_i^k}\right\|^2\right] = \mathbb{E}\left[\left\|\gamma \sum_{\ell=k-\delta_i^k}^{k-1} \bar{g}^\ell\right\|^2\right]$$

$$\leq 2\gamma^2 \mathbb{E}\left[\left\|\sum_{\ell=k-\delta_i^k}^{k-1} \frac{1}{n}\sum_{j=1}^n \nabla f_j\left(x^{\ell-\delta_j^\ell}\right)\right\|^2\right]$$

$$+ 2\gamma^2 \mathbb{E}\left[\left\|\sum_{\ell=k-\delta_i^k}^{k-1}\left(\bar{g}^\ell - \frac{1}{n}\sum_{j=1}^n \nabla f_j\left(x^{\ell-\delta_j^\ell}\right)\right)\right\|^2\right]$$

$$\leq 2\gamma^2\, \delta_i^k \underbrace{\sum_{\ell=k-\delta_i^k}^{k-1} \mathbb{E}\left[\left\|\frac{1}{n}\sum_{j=1}^n \nabla f_j\left(x^{\ell-\delta_j^\ell}\right)\right\|^2\right]}_{T_{ik}} + 2\gamma^2(\delta_i^k)^2\frac{\sigma^2}{Bn}\,.$$

In the last inequality, we used Jensen's inequality and Lemma E.2.

Next, we estimate the sum of $T_{ik}$

$$\sum_{k=0}^{K-1}\frac{1}{n}\sum_{i=1}^n T_{ik} = \sum_{k=0}^{K-1}\frac{1}{n}\sum_{i=1}^n \delta_i^k \sum_{\ell=k-\delta_i^k}^{k-1} \mathbb{E}\left[\left\|\frac{1}{n}\sum_{j=1}^n \nabla f_j\left(x^{\ell-\delta_j^\ell}\right)\right\|^2\right]$$

$$= \frac{1}{n}\sum_{i=1}^n\sum_{k=0}^{K-1}\delta_i^k \sum_{\ell=k-\delta_i^k}^{k-1} \mathbb{E}\left[\left\|\frac{1}{n}\sum_{j=1}^n \nabla f_j\left(x^{\ell-\delta_j^\ell}\right)\right\|^2\right]$$

$$\leq 2\sum_{i=1}^n\sum_{k=0}^{K-1}\sum_{\ell=k-\delta_i^k}^{k-1} \mathbb{E}\left[\left\|\frac{1}{n}\sum_{j=1}^n \nabla f_j\left(x^{\ell-\delta_j^\ell}\right)\right\|^2\right]$$

$$\leq 2\sum_{i=1}^n\delta_i^{\max}\sum_{k=0}^{K-1} \mathbb{E}\left[\left\|\frac{1}{n}\sum_{j=1}^n \nabla f_j\left(x^{k-\delta_j^k}\right)\right\|^2\right]$$

$$\leq 4n^2\sum_{k=0}^{K-1} \mathbb{E}\left[\left\|\frac{1}{n}\sum_{j=1}^n \nabla f_j\left(x^{k-\delta_j^k}\right)\right\|^2\right]\,.$$

In the first and last inequality, we used the bound $\delta_i^{\max} \leq 2n$ from Lemma 4.1. Finally, applying the stepsize condition $\gamma \leq 1/4nL$ yields the result. $\square$

# G PROOF OF THEOREM 5.3

Before proving the theorem, we first prove the following lemma.

## G.1 PROOF OF LEMMA 5.2

This lemma establishes a time bound for one round of iterations and, more importantly, a lower bound on $B$.

**Lemma 5.2.** Each block of $n$ consecutive iterations (each round) of Algorithm 1 takes at most $2\tau_n$ seconds. Moreover, we have

$$B \geq \frac{\tau_n}{2} \left( \frac{1}{n} \sum_{i=1}^{n} \tau_i \right)^{-1} = \frac{\tau_n}{2\tau_{\text{avg}}} .$$

*Proof.* The upper bound of $2\tau_n$ follows from the structure of the algorithm, which consists of two phases. In the first phase, the server waits until all workers complete at least one gradient computation, which takes at most $\tau_n$ seconds. In the second phase, the server applies the received gradients and waits for all ongoing computations to finish—which again takes at most $\tau_n$ seconds. Thus, the total time for $n$ iterations is bounded by $2\tau_n$.

We now prove the second part of the lemma. Recall that

$$B = \inf_{k \geq 0} B^k = \inf_{k \geq 0} \left( \frac{1}{n} \sum_{i=1}^{n} \frac{1}{b_i^k} \right)^{-1} \geq \left( \frac{1}{n} \sum_{i=1}^{n} \frac{1}{b_i} \right)^{-1} ,$$

where we define

$$b_i = \inf_{k \geq 0} b_i^k .$$

We are interested in the number of gradients stored in the table at iteration $k$. This count includes gradients computed during Phase 1 plus one additional gradient from Phase 2 (except for the worker that finished Phase 1 last).

Since every worker needs to compute at least one gradient during Phase 1, the slowest worker will take $\tau_n$ seconds to complete single gradient computation. During this $\tau_n$-second interval, faster workers $i < n$ may still be finishing gradients from the previous round's Phase 2 before starting their Phase 1 computations for the current round.

After completing any remaining Phase 2 work (which takes at most $\tau_i$ seconds), worker $i$ has at least $\tau_n - \tau_i$ seconds remaining to compute additional gradients for the current round's Phase 1. The number of gradients that worker $i$ can compute satisfies

$$b_i \geq \max \left\{ 1, \left\lceil \frac{\tau_n - \tau_i}{\tau_i} \right\rceil \right\} \geq \max \left\{ 1, \frac{\tau_n}{\tau_i} - 1 \right\} .$$

For workers $i$ where $\tau_n \geq 2\tau_i$, we have

$$\frac{\tau_n}{\tau_i} - 1 \geq \frac{\tau_n}{2\tau_i} ,$$

and hence

$$b_i \geq \frac{\tau_n}{2\tau_i} .$$

Plugging this bound into the expression for $B$ gives the claimed result. $\square$

We are now ready to prove Theorem 5.3.

**Theorem 5.3.** Let Assumptions 2, 3, and 1 hold. Let the stepsize in Ringleader ASGD (Algorithm 1) be $\gamma = \min \left\{ 1/8nL, \varepsilon B/10L\sigma^2 \right\}$. Then, under the *fixed computation model* (2), Ringleader ASGD achieves the optimal time complexity

$$\mathcal{O} \left( \frac{L\Delta}{\varepsilon} \left( \tau_n + \tau_{\text{avg}} \frac{\sigma^2}{n\varepsilon} \right) \right) .$$

*Proof.* We start with the iteration complexity from Theorem 5.1

$$K \geq \frac{32nL\Delta}{\varepsilon} + \frac{40L\Delta\sigma^2}{B\varepsilon^2} = \mathcal{O} \left( \frac{nL\Delta}{\varepsilon} \left( 1 + \frac{\sigma^2}{Bn\varepsilon} \right) \right) .$$

The time to do $n$ steps is at most $2\tau_n$ form Lemma 5.2. Then the time complexity is

$$2\tau_n \times \frac{K}{n} = \mathcal{O} \left( \tau_n \frac{L\Delta}{\varepsilon} \left( 1 + \frac{\sigma^2}{Bn\varepsilon} \right) \right) .$$

It remains to put $B \geq \tau_n/2\tau_{\text{avg}}$ from Lemma 5.2. $\square$

## H   TIME COMPLEXITY OF IA$^2$SGD

The iteration complexity of IA$^2$SGD (Wang et al., 2025) is

$$K = \mathcal{O}\left(\frac{\delta^{\max}L\Delta}{\varepsilon}\left(1+\frac{\sigma^2}{n\varepsilon}\right)\right).$$

We now analyze the corresponding wall-clock time under the *fixed computation model* (2). Since the algorithm performs an update whenever a single worker finishes a computation, we seek the minimal time $T$ such that

$$\sum_{i=1}^{n}\left\lfloor\frac{T}{\tau_i}\right\rfloor \geq K.$$

Observe that

$$\sum_{i=1}^{n}\frac{T}{\tau_i} \geq \sum_{i=1}^{n}\left\lfloor\frac{T}{\tau_i}\right\rfloor.$$

Hence, if we define $T'$ by

$$\sum_{i=1}^{n}\frac{T'}{\tau_i} = K,$$

then

$$T' = \left(\sum_{i=1}^{n}\frac{1}{\tau_i}\right)^{-1}K.$$

It follows that the minimal time $T$ is necessarily larger than $T'$.

It remains to bound $\delta^{\max}$. At initialization, all workers start computing their first gradients simultaneously. By the time the slowest worker completes its first gradient (at time $\tau_n$), the other workers may each have completed multiple gradients. In particular,

$$\delta^{\max} \geq \sum_{i=1}^{n}\left\lfloor\frac{\tau_n}{\tau_i}\right\rfloor.$$

Combining this with the iteration complexity bound, we obtain that the total runtime satisfies

$$T \geq c \times \frac{\tau_n L\Delta}{\varepsilon}\left(1+\frac{\sigma^2}{n\varepsilon}\right),$$

for some universal constant $c > 0$.

Note that the expression above should not be viewed as an exact upper bound on the runtime. It is better understood as a simplified estimate of $T$, which is sufficient for our purposes and provides a cleaner basis for comparison.

## I   IMPROVED MALENIA SGD

Malenia SGD has the following iteration complexity (Tyurin & Richtárik, 2024)

$$K \geq \frac{12\Delta L_f}{\varepsilon} + \frac{12\Delta L_f\sigma^2}{\varepsilon^2 nS},$$

where $S$ is a lower bound on the harmonic mean of the batch sizes, i.e.,

$$\left(\frac{1}{n}\sum_{i=1}^{n}\frac{1}{b_i^k}\right)^{-1} \geq S,$$

for all iterations $k$. In the original Malenia SGD analysis (Tyurin & Richtárik, 2024), this bound follows from the condition in (4), which fixes the same value of $S$ across all iterations.

In the fixed-time regime (2), however, this condition is no longer necessary. By adopting the same strategy as Ringleader ASGD (Algorithm 1)—namely, waiting for at least one gradient from each worker—we effectively replace $S$ with $B$ in the rate. This yields the following time complexity

$$\tau_n K = \frac{12\tau_n \Delta L_f}{\varepsilon}\left(1 + \frac{\sigma^2}{\varepsilon n B}\right).$$

Substituting the expression for $B$ from Lemma 5.2 and proceeding as in the proof of Theorem 5.3, we obtain the same overall time complexity as before—this time *without* requiring condition (4), which depends on knowing $\sigma$ and fixing $\varepsilon$ in advance.

Finally, note that this improvement is only valid in the fixed-time regime. In the setting with arbitrarily varying computation times, the same optimization cannot be applied, for the same reasons discussed for Ringleader ASGD in Appendix D.

## NOTE ON LLM USAGE

Large Language Models were used to help polish the writing of the manuscript. LLM usage did not affect the scientific content of the paper.

