# OpenReview forum: "Ringleader ASGD: The First Asynchronous SGD with Optimal Time Complexity under Data Heterogeneity"
_ICLR.cc/2026/Conference — ICLR 2026 Poster_

### Official Review · Reviewer_nJYq · 2025-10-16

**Soundness:** 3
**Presentation:** 2
**Contribution:** 2
**Rating:** 6
**Confidence:** 4

**Summary:**

The paper studies asynchronous optimization methods for smooth nonconvex functions while relaxing the similarity across workers’ data and considering heterogeneous computational times. The authors proposed a new asynchronous algorithm called Ringleader SGD, which they show to achieve the theoretically optimal rate in the regimes considered in the paper.

**Strengths:**

The paper proposed an asynchronous optimization method called Ringleader SGD. This method extends the previous method of Ringmaster SGD to heterogeneous settings. When $L = O(L_f)$, the method of Ringleader SGD is proven to match the algorithmic independent lower bound proven in (Tyurin & Richtarik, 2024). Despite the minimax optimal rate is already achieved by a synchronous method called Malenia SGD, Ringleader SGD is a truly asynchronous variant and does not discard any gradient works. Also, the new algorithm does not need to know the gradient variances and target accuracies.

I check the proofs of the main theoretical results in section 5. The proofs seem correct to me. However, I did not check the proofs of the additional results on time-varying delays.

**Weaknesses:**

The Malenia SGD is already optimal in the regime considered in this paper. Therefore, despite the arguments of “truly asynchronous method” and “no discarding work”, it is impossible to show any theoretical improvement over Malenia SGD in the same regime. I totally agree on this one, and this is not the weakness. *However,* I was wondering whether it is possible to show any improvement over Malenia SGD under certain cases. On the theory side, under additional restriction of data similarity, whether Ringleader SGD might be provably faster?

Since there isn’t any clear theoretical advantage and the scale of the numerical experiments provided in this paper is also not convincing enough, the truly non-incremental part of contribution seems to be a parameter-free method without knowing gradient variances and target accuracies in advance. Hence, I think this contribution alone is a bit limited.

**Questions:**

My only question is about the presentation of the results. Why didn’t the authors write down the algorithmic-independent lower bound from (Tyurin & Richtarik, 2024) explicitly in the paper? They claimed their algorithm is optimal throughout the paper, but the readers will have to download (Tyurin & Richtarik, 2024) and check the lower bound therein to confirm it. It would be great to cite the lower bound of $\Omega \left( \frac {L_f \Delta} {\epsilon} \left( \tau_n + \tau _{avg} \frac {\sigma^2} {n \epsilon} \right)  \right)$ as a separate theorem statement, or at least include it explicitly as a row in Table 1.

One suggestion is to present the results more modestly. While the authors keep claiming their methods are “optimal” (with the word “optimal” appearing for about 20 times in the paper), it should be noted that the optimality is only achieved when $L = O(L_f)$, and their method is not optimal in the general regime. The authors have mentioned this but only in the footnotes. I would recommend this restriction on “optimality” to be stated explicitly at least in the contribution and in the remarks below their main theorem. From the current presentation, it is very misleading in general, since their method might be even worse than Minibatch SGD in certain regimes. (Just to clarify, I think stating the exact restrictions helps to avoid vagueness, thus making the true contribution of this paper clearer.)

---

> ### Author Response · Authors · 2025-11-23
> **Answering Weaknesses and Questions**
>
> Thank you for taking the time to review our paper. Below, we respond to the weaknesses and questions you raised.
>
> > The Malenia SGD is already optimal in the regime considered in this paper. Therefore, despite the arguments of “truly asynchronous method” and “no discarding work”, it is impossible to show any theoretical improvement over Malenia SGD in the same regime. I totally agree on this one, and this is not the weakness. _However,_ I was wondering whether it is possible to show any improvement over Malenia SGD under certain cases. On the theory side, under additional restriction of data similarity, whether Ringleader SGD might be provably faster?
>
> We agree that if one imposes additional assumptions on data similarity, then it may be possible to obtain faster time-complexity guarantees than the optimal rate considered in our work. In such settings, one could potentially avoid storing a full gradient table and instead exploit the similarity structure to use gradients as they arrive, without having to wait for the slowest worker in every round, as in our algorithm.
>
> However, in that regime, it is not clear what the _correct_ optimal time complexity should be: the existing lower bound of Tyurin & Richtárik (2024) does not cover similarity assumptions. One would ideally also need matching lower bounds under those assumptions, which is a substantial additional project and outside the scope of this paper.
>
> We also note that many data-similarity assumptions are quite restrictive in practice. One of our goals here was to obtain an optimal-time asynchronous method **without** relying on such assumptions, which makes the theory cleaner and more generally applicable.
>
> > Since there isn’t any clear theoretical advantage and the scale of the numerical experiments provided in this paper is also not convincing enough, the truly non-incremental part of contribution seems to be a parameter-free method without knowing gradient variances and target accuracies in advance. Hence, I think this contribution alone is a bit limited.
>
> We respectfully disagree that the only non-incremental contribution is the parameter-free aspect. Even though we cannot improve on Malenia SGD’s rate (because it already matches the lower bound), our main goal is different: to show that an **asynchronous** method can _attain_ this optimal time complexity in the heterogeneous compute-time model.
>
> Until now, optimal time complexity in the heterogeneous-data case was attained only by a carefully synchronized method (Malenia SGD). The existence of a truly asynchronous method—one that does not discard completed gradients, works under heterogeneous worker speeds, and still matches the lower bound—is exactly the open question we set out to answer. From this perspective, the contribution is not a better rate over Malenia SGD, but the fact that **such an asynchronous optimal algorithm exists at all** in this setting.
>
> The parameter-free design and the experimental illustrations are additional advantages, but we view the main conceptual contribution as closing this long-standing gap: showing that one can have both asynchrony (with its practical benefits) and provable optimal time complexity under heterogeneity.
>
> ## Questions
>
> > My only question is about the presentation of the results. Why didn’t the authors write down the algorithmic-independent lower bound from (Tyurin & Richtarik, 2024) explicitly in the paper? They claimed their algorithm is optimal throughout the paper, but the readers will have to download (Tyurin & Richtarik, 2024) and check the lower bound therein to confirm it. It would be great to cite the lower bound as a separate theorem statement, or at least include it explicitly as a row in Table 1.
>
> Thank you for pointing this out. In the revised version, we now include the algorithm-independent lower bound from Tyurin & Richtárik (2024) explicitly as an additional row in Table 1, so that the optimality comparison is visible directly in our paper without needing to open the external reference.
>
> > One suggestion is to present the results more modestly. While the authors keep claiming their methods are “optimal” (with the word “optimal” appearing for about 20 times in the paper), it should be noted that the optimality is only achieved when $L = \mathcal{O}(L_f)$, and their method is not optimal in the general regime. The authors have mentioned this but only in the footnotes. I would recommend this restriction on “optimality” to be stated explicitly at least in the contribution and in the remarks below their main theorem. From the current presentation, it is very misleading in general, since their method might be even worse than Minibatch SGD in certain regimes.
>
> You are right that this restriction was too hidden in the first version (only in a footnote). In the revision, we now state it explicitly in the contribution paragraph, and again in Remark 5.4 after Theorem 5.3, as you suggested.

---

> > ### Comment · Reviewer_nJYq · 2025-11-24
> >
> > Thank the authors for the response and for making the revisions.
> >
> > I can see it is interesting to have an algorithm that is adaptive and does not discard gradient. My concern, however, remains. This algorithm does not get any theoretical benefit (even under additional assumption it is good to see some theoretical gain in certain regime); and the practical performance is tested on limited numerical cases. In general, I am not fully convinced by the impact of the method.
> >
> > Therefore, I will not increase my score (given that my original score is already positive).

---

### Official Review · Reviewer_M5AX · 2025-10-26

**Soundness:** 3
**Presentation:** 3
**Contribution:** 3
**Rating:** 6
**Confidence:** 2

**Summary:**

This paper proposes Ringleader ASGD, an asynchronous stochastic gradient descent (SGD) algorithm that claims to achieve optimal time complexity under both data and computation heterogeneity. The method integrates mechanisms from several prior works: it uses a gradient table similar to IA2SGD, employs delay control ideas from Ringmaster ASGD, and adopts a two-phase update structure inspired by Malenia SGD. Theoretical results show that, under a fixed computation-time model, the algorithm matches the known lower bound for parallel first-order stochastic methods, thereby closing a theoretical gap regarding the optimality of asynchronous SGD in heterogeneous settings.
﻿

I should note that I am **not** an expert in asynchronous or distributed optimization theory, so my ability to judge the novelty and technical rigor of the proofs is limited. From my reading, the paper appears clearly written and the analysis internally consistent, but I would encourage the area chair to give greater weight to the opinions of reviewers with deeper expertise in this area, particularly regarding the theoretical contributions and their significance.

**Strengths:**

1. **Strong theoretical contribution.**
The paper provides a rigorous theoretical analysis establishing that Ringleader ASGD achieves the *optimal time complexity* under heterogeneous data and computation speeds. This closes an open gap left by prior work which achieved optimality only in synchronous or homogeneous settings.
﻿
2. **Clear connection to prior literature.**
The authors carefully position their method relative to previous parallel SGD variants and articulate how each component contributes to achieving both asynchrony and optimality. The theoretical results are clearly stated and supported by well-organized proofs in the appendix.
﻿
3. **Conceptual clarity of the design.**
The proposed two-phase structure (gradient collection and sequential asynchronous updates) provides a clear mechanism to control gradient delays while avoiding idle workers or wasted computation. The algorithmic description is logically consistent and easy to follow.

**Weaknesses:**

The experimental section is limited and does not convincingly demonstrate the practical relevance of the proposed algorithm. Experiments are restricted to small datasets (MNIST, Fashion-MNIST) and a simple two-layer MLP, which do not reflect realistic distributed or federated learning scenarios.

**Questions:**

The experimental section is rather limited, focusing only on small-scale simulations with simple MLP models on MNIST and Fashion-MNIST. Could the authors provide more comprehensive empirical evaluations to better demonstrate the practical impact of the proposed method?

---

> ### Author Response · Authors · 2025-11-22
> **Response**
>
> We thank the reviewer for taking the time to read our paper and for the clear and positive summary of our work. We appreciate the constructive tone and the effort to accurately capture the main ideas and the positioning with respect to prior methods (IA2SGD, Ringmaster ASGD, Malenia SGD).
>
> We also appreciate the reviewer’s honest remark that they are not a specialist in asynchronous or distributed optimization theory. As you note, this naturally makes it harder to fully assess the theoretical novelty and technical depth of the results. Our main contribution, however, is precisely of this nature: the paper is predominantly **theoretical**, and it focuses on closing a **time-complexity gap that has remained open for more than a decade** in the asynchronous SGD literature, starting from the seminal work HOGWILD! ([Recht et al., 2011](https://proceedings.neurips.cc/paper/2011/file/218a0aefd1d1a4be65601cc6ddc1520e-Paper.pdf)) and its many follow-ups.
>
> Concretely, we show that under a heterogeneous compute-time model and under the heterogeneous data setting, Ringleader ASGD matches the known lower bound for parallel first-order stochastic methods, thereby resolving the question of whether an _optimal_ asynchronous SGD algorithm exists in this realistic heterogeneous setting. This is the core novelty and motivation of the work.
>
> Below, we address your weaknesses and questions.
>
> ---
> > The experimental section is limited and does not convincingly demonstrate the practical relevance of the proposed algorithm. Experiments are restricted to small datasets (MNIST, Fashion-MNIST) and a simple two-layer MLP, which do not reflect realistic distributed or federated learning scenarios.
>
> We agree that the experimental section is limited in scale and does not aim to simulate a full, production-level distributed or federated learning setup. This is by design. The goal of the paper is **not** to introduce yet another practical training recipe, but to resolve a theoretical question about the _time complexity_ of asynchronous SGD under heterogeneity.
>
> The experiments we included are intended as **illustrative sanity checks** of our compute-time model and the qualitative behaviour of Ringleader ASGD, rather than as the main contribution. The core results of the paper---the optimal time-complexity guarantees and the closure of a long-standing theoretical gap---stand independently of these particular experiments, and the paper would remain meaningful even in their absence.
>
> ---
> > The experimental section is rather limited, focusing only on small-scale simulations with simple MLP models on MNIST and Fashion-MNIST. Could the authors provide more comprehensive empirical evaluations to better demonstrate the practical impact of the proposed method?
>
> We fully agree that more extensive empirical studies (larger models, larger-scale systems) are interesting and important, but we view them as natural follow-up work building on the theoretical foundation established here, rather than as the central criterion for evaluating this submission.

---

> > ### Comment · Reviewer_M5AX · 2025-11-24
> >
> > I thank the authors for their response to my review. I am satisfied to keep my score where it is.

---

### Official Review · Reviewer_kQKe · 2025-11-01

**Soundness:** 3
**Presentation:** 2
**Contribution:** 3
**Rating:** 4
**Confidence:** 4

**Summary:**

This paper introduces Ringleader ASGD, a stochastic gradient descent method designed to achieve optimal time complexity in distributed nonconvex optimization under data heterogeneity. The work addresses a fundamental gap in asynchronous optimization theory by proposing a method that avoids restrictive similarity assumptions across workers' data distributions while matching established lower bounds.

**Strengths:**

The paper provides a rigorous analysis of Ringleader ASGD, proving optimal time complexity.

**Weaknesses:**

1. The presence of the initial gradient collection phase (Phase 1) introduces a dependency whereby the failure of any single worker node halts the entire algorithmic process. Consequently, Ringleader ASGD cannot be classified as purely asynchronous. Its core design represents a hybrid methodology that integrates elements of both synchronous and asynchronous paradigms.

2. Each iteration of Ringleader ASGD may require numerous gradient computations. The paper only provides both iteration complexity and time complexity. Could you analyze its computational complexity and compare it with other methods?

3. In my opinion, academic writing should adhere to fundamental structural conventions. The experimental section and conclusions should be placed within the main body of the text rather than in an appendix.

4. The experimental scope could be strengthened by evaluating the method on more challenging benchmarks. Expanding the experiments to include larger-scale datasets (e.g., ImageNet, CIFAR-100) and more complex model architectures would help better validate the robustness and scalability of the proposed approach.

**Questions:**

Refer to Weaknesses.

---

> ### Author Response · Authors · 2025-11-22
> **Addressing the Weaknesses 1**
>
> We thank the reviewer for the detailed and thoughtful comments. We have carefully gone through all the points you raised and respond to them one by one below. If any aspects remain unclear after reading our responses, we would be very happy to continue the discussion during the rebuttal phase. We hope that these clarifications will address your concerns and lead you to reconsider your overall assessment of the paper.
>
> ---
>
> > 1. The presence of the initial gradient collection phase (Phase 1) introduces a dependency whereby the failure of any single worker node halts the entire algorithmic process.
>
> In our setting, we aim to minimize
> $$
> f(x) = \frac{1}{n}\sum_{i=1}^n f_i(x)
> $$
> without any similarity assumption between clients. If a single worker fails *permanently*, then, in general, no method can solve this problem to an arbitrarily small stationarity level: the contribution of the missing $f_i$ cannot be recovered, so $\|\nabla f(x)\|$ cannot be guaranteed to go below an arbitrary $\varepsilon$.
>
> On the other hand, **temporary** slowdowns or outages of a worker do *not* halt the entire process in Ringleader ASGD. While one worker is delayed, all other workers keep computing gradients, and the server keeps accumulating them. Thus, all available workers are doing useful computation all the time; the algorithm does not globally “stop” while waiting.
>
> ---
>
> > Consequently, Ringleader ASGD cannot be classified as purely asynchronous. Its core design represents a hybrid methodology that integrates elements of both synchronous and asynchronous paradigms.
>
> We agree that Ringleader ASGD combines asynchronous execution with a coordination mechanism, and in that sense it has both “async” and “sync-like” aspects. The key point, however, is **what we mean by “asynchronous” in the standard optimization sense**, and why Phase 1 is needed at all.
>
> ## Why some coordination is necessary
>
> The fundamental difficulty in asynchronous methods is the presence of **delays**: updates applied using stale gradients. In the homogeneous-data setting, it is known that if delays are not properly controlled, asynchronous methods fail to achieve optimal time complexity ([Maranjyan et al., 2025d](https://proceedings.mlr.press/v267/maranjyan25b.html)). Exactly the same phenomenon appears in the heterogeneous-data setting: IA-SGD is a naive asynchronous method in the usual sense, but because it does not control delays, it is provably *not* time-optimal in our model.
>
> To reach the optimal rate under heterogeneous data, we must control the delays $\delta_i^k$ while still allowing workers to compute continuously. The two-phase structure of Ringleader (Phase 1 + Phase 2) is precisely what ensures a **uniform bound** on delays (Lemma 3.1 in our paper) without discarding any gradients. This kind of structured coordination is therefore not an arbitrary design choice but a necessary ingredient for optimality.
>
> ## Synchronous vs asynchronous: our definition
>
> In our terminology, the distinction is:
>
> - **Synchronous:** every update uses gradients evaluated at the **current** server model $x^k$. That is,
>   $$
>     x^{k+1} = x^k - \gamma^k \sum_{i=1}^n w_i^k \sum_{j=1}^{b_i^k} \nabla f_i(x^k;\xi_i^{k,j}),
>   $$
>   so all gradients in the update come from the same model state. This forces either global waiting or discarding of late gradients.
>
> - **Asynchronous:** updates may use gradients evaluated at **stale** model copies. Formally, we allow
>   $$
>     x^{k+1} = x^k - \gamma^k \sum_{i=1}^n w_i^k \sum_{j=1}^{b_i^k} \nabla f_i(x^{k - \delta_i^k};\xi_i^{k-\delta_i^k, j}),
>   $$
>   where $\delta_i^k \ge 0$ is the delay for worker $i$ at iteration $k$. When some $\delta_i^k > 0$, the update uses a mixture of gradients computed at different model states, and the server cannot be viewed as operating in lockstep with all workers.
>
> The whole *point* of calling a method “asynchronous” is that its update rule and theory are explicitly designed to handle such delays $\delta_i^k > 0$, rather than assuming they are always zero.
>
> Ringleader ASGD fits exactly into this asynchronous template: in both phases, the server updates using gradients evaluated at possibly stale iterates $x^{k-\delta_i^k}$, and we explicitly prove a delay bound $\delta_i^k \le 2n-2$. Workers compute continuously and do not wait for a global barrier at every update. In this sense, Ringleader is an asynchronous algorithm with **bounded delays**, not a fully synchronous one.
>
> Phase 1 introduces *round-level coordination* (we wait until each worker has contributed at least one gradient), but the *updates themselves* are asynchronous: they mix gradients computed at different times and different model states, and workers keep computing while the server updates.

---

> > ### Author Response · Authors · 2025-11-22
> > **Addressing the Weaknesses 2**
> >
> > > 2. Each iteration of Ringleader ASGD may require numerous gradient computations. The paper only provides both iteration complexity and time complexity. Could you analyze its computational complexity and compare it with other methods?
> >
> > All three methods we compare—IA-SGD, Malenia SGD, and Ringleader ASGD—are designed so that **workers are always busy**: they compute gradients continuously and never idle on purpose. This implies that **time complexity and computational complexity are equivalent up to constant factors**: if an algorithm is faster in wall-clock time with all workers fully utilized, it also uses fewer total gradients (up to constants), and vice versa.
> >
> > We can also see this explicitly by computing the computational complexity (total number of gradients) for Ringleader ASGD and IA-SGD, without going through the time complexity.
> >
> > ---
> >
> > ## Computational complexity of Ringleader ASGD
> >
> > From Theorem 5.1, the iteration complexity of Ringleader ASGD is
> > $$
> > K_{\text{Ring}}
> > = \mathcal{O} \left(
> >     \frac{nL\Delta}{\varepsilon} \left(1 + \frac{\sigma^2}{B n \varepsilon}\right)
> >   \right),
> > $$
> > where
> > $$
> > B = \inf_{k \ge 0} \left( \frac{1}{n} \sum_{i=1}^n \frac{1}{b_i^k} \right)^{-1} \ge 1,
> > $$
> > and $b_i^k$ is the number of gradients from worker $i$ used in the update at iteration $k$.
> >
> > From the structure of the algorithm we have the following bound on $b_i^k$.
> > During Phase 1, the slowest worker (with time $\tau_n$) can compute at most $2$ gradients before the phase ends, and then it contributes at most one more gradient in Phase 2 for its update in that round. Faster workers can compute proportionally more, so
> > $$
> > b_i^k \le  2\frac{\tau_n}{\tau_i} + 1 \le 3 \frac{\tau_n}{\tau_i}.
> > $$
> > Hence
> > $$
> > \sum_{i=1}^n b_i^k \le 3\tau_n \sum_{i=1}^n \frac{1}{\tau_i}.
> > $$
> >
> > Let $G_{\text{Ring} }$ denote the total number of gradients computed up to iteration $K_{\text{Ring}}$.
> > Because $n$ updates happen in phase 2, we get
> > $$
> > G_{\text{Ring}} \le \frac{K_{\text{Ring}} }{n} 3\tau_n \sum_{i=1}^n \frac{1}{\tau_i}
> > $$
> > Substituting the expression for $K_{\text{Ring}}$, we obtain
> > $$
> > G_{\text{Ring}} = \mathcal{O} \left(
> >     \tau_n \sum_{i=1}^n \frac{1}{\tau_i}
> >     \frac{L\Delta}{\varepsilon} \left(1 + \frac{\sigma^2}{B n \varepsilon}\right)
> >   \right).
> > $$
> >
> > ---
> >
> > ## Computational complexity of IA-SGD
> >
> > The iteration complexity of IA-SGD ([Wang et al., 2025](https://openreview.net/forum?id=m3x4kDbYAK)) can be written as
> > $$
> > K_{\text{IA}}
> > = \mathcal{O}\left(
> >     \delta^{\max} \frac{L\Delta}{\varepsilon}\left(
> >       1 + \frac{\sigma^2}{n\varepsilon}
> >     \right)
> >   \right),
> > $$
> > where $\delta^{\max}$ is the maximum delay across all workers and all iterations. Each iteration of IA-SGD uses exactly **one** stochastic gradient, so the computational complexity is simply
> > $$
> > G_{\text{IA}} = K_{\text{IA}}.
> > $$
> > In the fixed computation-time model, we can lower-bound how large $\delta^{\max}$ can be in heterogeneous regimes. By the time the slowest worker finishes its **first** gradient (at time $\tau_n$), worker $i$ may have computed approximately $\big\lfloor \tau_n / \tau_i \big\rfloor$ gradients. Hence, there exists a gradient from the slowest worker whose delay is at least
> > $$
> > \delta^{\max}  \ge \sum_{i=1}^n \left\lfloor \frac{\tau_n}{\tau_i} \right\rfloor \approx \tau_n \sum_{i=1}^n \frac{1}{\tau_i}.
> > $$
> > Plugging this into the expression for $K_{\text{IA}}$ shows that in such heterogeneous scenarios,
> > $$
> > G_{\text{IA}}
> > =
> > \Omega \left(
> >     \tau_n \sum_{i=1}^n \frac{1}{\tau_i}
> >      \frac{L\Delta}{\varepsilon}\left(
> >       1 + \frac{\sigma^2}{n\varepsilon}
> >     \right)
> >   \right).
> > $$
> >
> > ---
> >
> > ## Comparison
> >
> > Comparing the two:
> > - Ringleader ASGD has
> >   $$
> > G_{\text{Ring}} = \mathcal{O}\left(
> >       \tau_n \sum_{i=1}^n \frac{1}{\tau_i}
> >       \frac{L\Delta}{\varepsilon} \left(1 + \frac{\sigma^2}{B n \varepsilon}\right)
> >     \right),
> >   $$
> >   with the variance term scaled by $B n \varepsilon$, where $B \ge 1$.
> >
> > - IA-SGD can require at least
> >   $$
> >   G_{\text{IA}}
> >   = \Omega\left(
> >       \tau_n \sum_{i=1}^n \frac{1}{\tau_i}
> >        \frac{L\Delta}{\varepsilon}\left(
> >         1 + \frac{\sigma^2}{n\varepsilon}
> >       \right)
> >     \right),
> >   $$
> >   with the variance term scaled by $n\varepsilon$.
> >
> > Thus, both in terms of **time** and in terms of **total gradient computations**, Ringleader ASGD has strictly better guarantees.
> >
> > With a similar argument, it can be shown that the computational complexity of RIngleader ASGD and Malenia SGD is the same up to a constant.

---

> > > ### Author Response · Authors · 2025-11-22
> > > **Addressing the Weaknesses 3**
> > >
> > > > 3. In my opinion, academic writing should adhere to fundamental structural conventions. The experimental section and conclusions should be placed within the main body of the text rather than in an appendix.
> > >
> > > Thank you for this suggestion.
> > >
> > > We have uploaded a new version in which the **conclusion** is now part of the main body. In the initial submission it was placed in the appendix purely due to space constraints.
> > >
> > > Regarding the **experiments**, we intentionally keep the detailed experimental section in the appendix. This work is primarily theoretical: the main contribution is resolving an open optimality question for asynchronous SGD. The experiments are meant to *support* the theory rather than drive the paper, and the current structure reflects that emphasis. We believe the paper should stand on its theoretical merits even without experiments, and the appendix format allows us to include the empirical validation without compressing or omitting key parts of the analysis in the main text.
> > >
> > >
> > > ----
> > >
> > >
> > > > 4. The experimental scope could be strengthened by evaluating the method on more challenging benchmarks. Expanding the experiments to include larger-scale datasets (e.g., ImageNet, CIFAR-100) and more complex model architectures would help better validate the robustness and scalability of the proposed approach.
> > >
> > > As mentioned above, this paper is **primarily theoretical**: the main contribution is to resolve an optimality question for asynchronous SGD in the heterogeneous-data setting. The role of the experiments here is deliberately modest—they serve as a sanity check that the behavior observed in practice is consistent with the theory, not as the main pillar of the contribution.
> > >
> > > Extending the experimental section to large-scale benchmarks would require substantial additional systems work, which we view as valuable but **orthogonal** to the core theoretical message of the paper. We see this as a natural and important direction for follow-up work, but we believe the current experimental scope is appropriate for a theory-focused contribution and sufficient to support our claims.

---

> > ### Comment · Reviewer_kQKe · 2025-11-28
> >
> > I agree with the author’s viewpoint that there is an inherent contradiction between data heterogeneity and computational performance heterogeneity in asynchronous algorithms. It is impossible to resolve the issue of data heterogeneity while allowing arbitrary computational performance heterogeneity for asynchronous algorithms. By enforcing bounded delays, this paper makes a significant attempt to address this challenge. Therefore, I intend to raise my score to 6.

---

> > > ### Author Response · Authors · 2025-11-28
> > >
> > > Thank you for your willingness to increase the score. Could you please also update the score in the initial review, as it still appears unchanged on our side? We also invite you to consider our responses to the other weaknesses you raised; we believe they address your concerns and could justify a higher score, if you find them convincing. If not, we would be very happy to keep engaging with you, as we believe we can resolve all remaining concerns.

---

> > > > ### Author Response · Authors · 2025-11-29
> > > > **Note on reviewer’s intended score change**
> > > >
> > > > It seems that the reviewers can no longer adjust their scores or comment further due to the recent change related to the OpenReview bug.
> > > >
> > > > Dear AC, please note that Reviewer kQKe had expressed a willingness to increase their score, and we were hoping to continue the discussion and potentially address their concerns further. We kindly ask that you take this into account when making your decision on the paper.

---

### Official Review · Reviewer_rk1S · 2025-11-01

**Soundness:** 3
**Presentation:** 3
**Contribution:** 3
**Rating:** 4
**Confidence:** 4

**Summary:**

This paper studies the task of finding an approximate stationary point for the average of smooth and non-convex objectives in a distributed optimization framework. It proposes and analyzes a new asynchronous optimization algorithm, Ringleader ASGD. The primary setting the paper discusses is that of fixed but heterogeneous computation speeds on different clients. In the regime where communication times to and from the server are negligible, the algorithm is shown to have optimal time complexity, up to a difference in the second-order smoothness constants across the machines.

In the appendix, the analysis is also extended to other, more general asynchronous models, along with experiments that demonstrate Ringleader SGD offers performance competetive to other baseline algorithms.

**Strengths:**

The paper is well-written, the setting is rigorously defined, and the baseline algorithms and related works are discussed clearly and sufficiently. The key theoretical result of the paper is stated clearly, and both the build-up to the algorithms and their proof sketch provide helpful intuition for the problem. Overall, it is straightforward to follow the paper's contribution. The experiments offered in the appendix also provide insight into the algorithm's actual performance.

**Weaknesses:**

My main criticism of the paper and the related literature is that it seems unreasonable to ignore communication costs, given that they are well known to often comprise one of the most significant chunks of overall time complexity in practice, in both synchronous and asynchronous workloads. Within the scope of the assumptions made in the paper, all the results and conclusions are valid; however, I believe that emphasizing optimality in this arguably unrealistic computation-communication model is potentially misleading. I list some explicit questions about this below.

I do not expect the authors to consider communication costs in this work, but I want to see a thorough discussion of the limitations of this setting before I am convinced to raise my score.

Similarly, I am also a little concerned about the claim that the work does not consider any heterogeneity assumptions, because Assumption 2 appears to be one. Specifically, the assumption appears to be related to the second-order heterogeneity assumption, which has been considered in the [non-convex](https://openreview.net/forum?id=SNElc7QmMDe) and [convex](https://arxiv.org/abs/2405.11667) settings.

**Questions:**

1. The authors discuss the issue of non-negligible communication costs (arguably the most realistic setting) only once while mentioning Tyurin et al. (2024b). On the other hand, multiple places in the paper, including the title, discuss optimal time complexity. This is a bit misleading. I urge the authors to dedicate a whole section to communication costs, discussing any known results in the setting with communication costs, their own algorithm and its guarantees in that setting, and potential sources of sub-optimality. In the experiments, it would be beneficial to simulate communication costs and demonstrate how the proposed algorithm performs compared to other baselines, as communication forms a larger portion of the overall physical time. Given that the improvements with respect to IA-SGD are not drastic, I am curious how these different algorithms compare when communication costs increase. Intuitively, since IA-SGD has lower overhead per communication (because buffers and other states do not need to be maintained/updated on the server), one would expect it to improve upon ringleader SGD when communication costs increase/overhead per communication increases. This can be simulated by changing model size or by increasing parallelization, which would force the server to "wait out" any clients trying to communicate.

2. It would also be good to discuss as well as implement a variant of minibatch SGD with different mini-batch sizes on each machine, which is another natural synchronous baseline in the face of different compute speeds. In a sense, this is the simplest synchronous algorithm, not the naive mini-batch discussed in Section 3. When communication costs are high, I would expect this algorithm to outperform both IA-SGD and ringleader SGD, as it requires fewer locks (a single lock per communication round between all devices). In the setup discussed in the previous bullet point, as the authors vary the communication cost, this should intuitively beat all baselines at some point.

The last two points highlight the age-old heuristic in distributed optimization: "Is the precious compute on the server better spent in compressing/implementing asynchrony/de-biasing or running more synchronous updates for a fixed time budget?" Naturally, this is an empirical question, and the role of the theory should be to help weed out different regimes where different choices make sense.

3. The discussion around Assumption 2 and the need for data similarity assumptions in this paper is a bit unclear to me. To me, it appears that in the non-trivial regime where $L^2$ is smaller than the average smoothness-squared across the machines, it essentially amounts to a data heterogeneity assumption. For instance, in the synchronous setting where Assumption 2 is only required on $x = \frac{i}{n}\sum_{i\in[n]}y_i$, then it is akin to a second-order data heterogeneity assumption, which several works have considered for analysing local update algorithms ([1](https://openreview.net/forum?id=SNElc7QmMDe), [2](https://arxiv.org/abs/2008.03606), [3](https://arxiv.org/abs/2102.03198), [4](https://arxiv.org/abs/2405.11667), [5](https://arxiv.org/abs/1910.06378)). I understand that if the authors assume all clients are smooth, it is in some sense not restrictive to think of $L$ as just a smoothness constant; however, I would like to see a discussion of the second-order heterogeneity literature. More specifically, can the analysis in this paper be extended to the setting with a bounded second-order heterogeneity constant? Hence, the time-complexity decouples between a heterogeneity term, and another that depends on $L_f$? As a motivation, in the quadratic setting, instead of the right-hand side of Assumption 2 with just L one can write an upper bound of, $\frac{2}{n}\sum_{i\in[n]}\left(L_f^2\|\|x-y_i\|\|_2^2 + \tau^2 \|\| y_i - \bar y\|\|_2^2\right),$ where $\bar y$ is the average of $y_i$'s and $\tau$ is the second-order heterogeneity parameter. It is unclear whether this alone will lead to an analysis in the asynchronous setting, but it at least highlights that the constant $L$ encapsulates both $L_f$ and a data-heterogeneity effect.

---

> ### Author Response · Authors · 2025-11-20
> **Addressing the Weaknesses 1**
>
> We thank the reviewer for taking the time to provide a detailed and thoughtful review. We have revised the paper based on your comments, including adding a new section that clarifies our modeling assumptions and the role of communication. Below, we address your concerns and hope that these clarifications will alleviate them and motivate an updated score.
>
> > My main criticism of the paper and the related literature is that it seems unreasonable to ignore communication costs, given that they are well known to often comprise one of the most significant chunks of overall time complexity in practice, in both synchronous and asynchronous workloads. Within the scope of the assumptions made in the paper, all the results and conclusions are valid; however, I believe that emphasizing optimality in this arguably unrealistic computation-communication model is potentially misleading. I list some explicit questions about this below.
>
> > I do not expect the authors to consider communication costs in this work, but I want to see a thorough discussion of the limitations of this setting before I am convinced to raise my score.
>
> We fully agree that communication is a major bottleneck in practical distributed systems, both in synchronous and asynchronous regimes. Our paper does *not* claim otherwise. However, the goal of this work is to answer a *different*, very specific theoretical question that has remained unresolved for over a decade:
>
> *Can asynchronous SGD solve the problem it is designed to solve—eliminating waiting caused by heterogeneous or straggling workers—and can it do so optimally?*
>
> To make this question mathematically well-defined, one must isolate the phenomenon that asynchrony is intended to address. We follow the classical “computation-only” model used in essentially all prior work on asynchronous optimization ([Mishchenko et al., 2022](https://proceedings.neurips.cc/paper_files/paper/2022/file/029df12a9363313c3e41047844ecad94-Paper-Conference.pdf); [Koloskova et al., 2022](https://proceedings.neurips.cc/paper_files/paper/2022/file/6db3ea527f53682657b3d6b02a841340-Paper-Conference.pdf); [Tyurin & Richtárik, 2024](https://proceedings.neurips.cc/paper_files/paper/2023/file/3563abb1040f4e150f4242a7282cd1ec-Paper-Conference.pdf); [Maranjyan et al., 2025d](https://proceedings.mlr.press/v267/maranjyan25b.html)), where communication is not modeled explicitly. This is also the model in which the lower bound of [Tyurin & Richtárik, 2024](https://proceedings.neurips.cc/paper_files/paper/2023/file/3563abb1040f4e150f4242a7282cd1ec-Paper-Conference.pdf) is derived. Our optimality claims are always with respect to this model.
>
> Below, we explain why this model is appropriate for our contribution and how it relates to communication.

---

> > ### Author Response · Authors · 2025-11-20
> > **Addressing the Weaknesses 2**
> >
> > ## What asynchrony is actually designed to solve
> >
> > Asynchrony was introduced to address a specific bottleneck: **waiting**. In a parallel or distributed system with heterogeneous or unstable workers, synchronous methods suffer from heterogeneous computation speeds across devices, hardware failures, or network issues that delay some workers. In all these cases, the slowest worker determines the pace of synchronous updates, and faster workers sit idle, waiting. The goal of asynchronous methods is:
> >
> > *To eliminate this idle time by allowing fast workers to continue contributing, even while slow or delayed workers are catching up.*
> >
> > In other words, asynchrony is a mechanism for reducing **waiting time**, not **communication time**. A worker whose gradient arrives late because it is still computing and a worker whose gradient arrives late because it is still transmitting over the network look identical to the server: in both cases, the gradient is delayed. Asynchrony ensures that such delays—regardless of whether they stem from computation, hardware, or communication—do not block the progress of other workers.
> >
> > ## Even in the simplest setting (no communication), optimal asynchronous SGD was unknown
> >
> > An important point is that, **even in the simplest possible setting**, the question of whether asynchrony helps was not understood: homogeneous data, no communication cost.
> >
> > Tyurin & Richtárik, 2024 proved a lower bound on time complexity for all parallel first-order methods in this model. They then constructed synchronous algorithms (Rennala SGD and Malenia SGD) that *achieve* this lower bound, implying they are optimal in that sense. However, these methods are synchronous, and **no asynchronous method matched this lower bound**. As a result, all known asynchronous methods were **provably worse** than the optimal synchronous ones, even though asynchrony is meant to help in exactly this distributed setting.
> >
> > This negative result raises a fundamental concern:
> >
> > *Does asynchrony actually help at all, even in the idealized model without communication cost?*
> >
> > Only very recently, [Maranjyan et al., 2025d](https://proceedings.mlr.press/v267/maranjyan25b.html) resolved this question in the *homogeneous-data* setting by designing an asynchronous method (Ringmaster ASGD) that *matches the optimal time complexity* of the best synchronous method.
> >
> > Our work extends this understanding to the more challenging **heterogeneous-data** setting, where each worker has their own data distribution. Thus, our contribution is to show that **asynchrony can, in principle, solve the problem it is designed for and do so optimally**, even in the presence of heterogeneous data—something that was not known before.
> >
> > ## Communication efficiency is orthogonal and requires additional mechanisms
> >
> > Reducing *communication cost* itself requires different tools: gradient compression and quantization, sparsification, local updates (e.g. FedAvg-style methods), communication skipping or batching, etc. Asynchrony alone does not change message sizes, total bytes sent, or physical network latency.
> >
> > To the best of our knowledge, there is only one work that systematically studies time complexity while explicitly accounting for communication time. [Tyurin et al. (2024b)](https://papers.nips.cc/paper_files/paper/2024/file/06e2dd57e90a736a5a1fd3bb2bf95c6c-Paper-Conference.pdf) analyze time complexity in the homogeneous-data setting for parallel methods **with non-negligible communication time**. To achieve optimality in that setting, they use **gradient compression** to mitigate the communication bottleneck and combine it with a carefully designed minibatch SGD scheme that has asynchronous components and is closely related to the Rennala SGD algorithm. This is exactly what we mean by an _orthogonal_ direction: one starts from an algorithm that is optimal in the computation-only model and then adds additional mechanisms (such as compression) to make it optimal also when communication time is taken into account.
> >
> > We view our work as complementary to this line: one first understands the optimal behavior in the computation-only model (with and without asynchrony), and then combines this with communication-efficient mechanisms in richer system models. For the reasons above, we adopt the standard computation-only model used in essentially all theoretical work on asynchronous SGD, which is also the model behind the lower bounds we match. Our optimality statements are always relative to this shared and well-established abstraction. Studying asynchrony under explicit communication cost—where it must interact with compression, local updates, or buffering—requires new lower bounds and a different theoretical framework, and is beyond the scope of this work.
> >
> > In response to this comment, we have added a dedicated appendix section, **“The Computation-Only Model and the Role of Asynchrony”** (Section C).

---

> ### Author Response · Authors · 2025-11-20
> **Addressing the Weaknesses 3**
>
> > Similarly, I am also a little concerned about the claim that the work does not consider any heterogeneity assumptions, because Assumption 2 appears to be one. Specifically, the assumption appears to be related to the second-order heterogeneity assumption, which has been considered in the non-convex and convex settings.
>
> We appreciate this observation and agree that Assumption 2 is related in spirit to “second-order heterogeneity” conditions. Our claim about not using similarity assumptions is meant in the standard sense used in the asynchronous optimization literature. Below, we clarify what we mean and how Assumption 2 fits in.
>
> ## Prior work: smoothness + bounded heterogeneity
>
> Classical asynchronous methods with heterogeneous data (e.g., [Koloskova et al., 2022](https://proceedings.neurips.cc/paper_files/paper/2022/file/6db3ea527f53682657b3d6b02a841340-Paper-Conference.pdf); [Nguyen et al., 2022](https://proceedings.mlr.press/v151/nguyen22b/nguyen22b.pdf); [Mishchenko et al., 2022](https://proceedings.neurips.cc/paper_files/paper/2022/file/029df12a9363313c3e41047844ecad94-Paper-Conference.pdf)) typically impose *two* types of assumptions:
> 1. Each local objective is smooth: $f_i$ is $L_{f_i}$–smooth.
> 2. There is an explicit *bounded heterogeneity/similarity* condition, for example
>    $$
> 	   \| \| \nabla f_i(x) - \nabla f(x) \| \|^2 \le \zeta^2 \quad \text{for all } x.
>    $$
> The second type is what we refer to as a *similarity assumption*: an extra structural constraint that quantitatively bounds how far $\nabla f_i(x)$ is allowed to deviate from $\nabla f(x)$. This introduces a heterogeneity constant $\zeta^2$ that appears explicitly in the rate and degrades performance as data become more heterogeneous.
>
> The reason these works *need* such a condition is that their algorithms update using **one local gradient at a time**. At each iteration, the analysis has to relate a single $\nabla f_i(x)$ to the global $\nabla f(x)$; without a bound like the one above, there is no way to control progress for $f$.
>
> ## Why we do not refer to Assumption 2 as a similarity assumption
>
> In our case, the situation is different at the algorithmic level: Ringleader ASGD is designed so that **every update uses information from all workers** via the gradient table. The server always updates with an average of gradients coming from *all* clients (at delayed points), rather than a single $\nabla f_i$.
>
> Because of this design, our analysis does **not** need any per-iteration bound of the form $\| \| \nabla f_i(x) - \nabla f(x) \| \|$, and we never introduce a separate heterogeneity constant like $\zeta$ that appears in the rate. We only require a single constant $L$ that plays a smoothness-like role and captures how the averaged gradient behaves when gradients are evaluated at different (delayed) points due to asynchrony.
>
> Some form of smoothness is unavoidable in any first-order analysis: one always has to relate gradients at different points. In our setting, asynchrony naturally leads to gradients evaluated at *different* iterates across workers, and Assumption 2 is exactly the way this requirement appears in our analysis. It is therefore a smoothness-type condition tailored to the asynchronous setting, rather than an extra “bounded heterogeneity penalty” on top of smoothness.
>
> ## How Assumption 2 relates to smoothness and similarity
>
> As we show in Lemma E.1, the constant $L$ in Assumption 2 satisfies
> $$
> L_f \le L \le \sqrt{\frac{1}{n}\sum_{i=1}^n L_{f_i}^2} \le L_{\max}.
> $$
> In particular, in the perfectly homogeneous / similarity case $f_i \equiv f$, Assumption 2 reduces to the standard smoothness of $f$, and then $L = L_f$. Thus, as the $f_i$ become “more similar”, $L$ can be closer to $L_f$; as they differ more, $L$ may approach $\sqrt{\frac{1}{n}\sum_i L_{f_i}^2}$.
>
> So we fully agree that there is a connection to similarity in the sense that “more similar” local functions may yield a smaller effective constant $L$.
> However, the key conceptual distinction for us is that
> - prior asynchronous methods require an *extra* heterogeneity/similarity assumption (with a separate constant like $\zeta$) because they cannot be analyzed otherwise, whereas
> - our method can be analyzed **without** such a bounded-discrepancy condition, since each update aggregates all workers’ information, and the only additional structure we need beyond standard smoothness is the $L$–type relation above, which naturally arises from dealing with gradients at different points in an asynchronous setting.

---

> > ### Author Response · Authors · 2025-11-20
> > **Addressing the Questions 1**
> >
> > > 1. The authors discuss the issue of non-negligible communication costs (arguably the most realistic setting) only once while mentioning Tyurin et al. (2024b). On the other hand, multiple places in the paper, including the title, discuss optimal time complexity. This is a bit misleading. I urge the authors to dedicate a whole section to communication costs, discussing any known results in the setting with communication costs, their own algorithm and its guarantees in that setting, and potential sources of sub-optimality. In the experiments, it would be beneficial to simulate communication costs and demonstrate how the proposed algorithm performs compared to other baselines, as communication forms a larger portion of the overall physical time.
> >
> > We have addressed this point in detail in the *Addressing the Weaknesses 1 & 2* comment above and have added a dedicated appendix section, **“The Computation-Only Model and the Role of Asynchrony”** (Section C), which discusses this point.
> >
> > ---
> > > Given that the improvements with respect to IA-SGD are not drastic, I am curious how these different algorithms compare when communication costs increase. Intuitively, since IA-SGD has lower overhead per communication (because buffers and other states do not need to be maintained/updated on the server), one would expect it to improve upon ringleader SGD when communication costs increase/overhead per communication increases. This can be simulated by changing model size or by increasing parallelization, which would force the server to "wait out" any clients trying to communicate.
> >
> > We thank the reviewer for this comment, but we respectfully disagree with the intuition that IA-SGD becomes preferable once communication costs are taken into account.
> >
> > As presented in the paper, Ringleader ASGD is described in the computation-only model, where workers send a gradient to the server as soon as it is computed. We chose this formulation because, in our theoretical setting, communication takes zero time and this “send-every-gradient” view makes the algorithm much easier to present and understand. In a realistic system where communication is costly, one would naturally implement Ringleader ASGD differently.
> >
> > A communication-efficient implementation of Ringleader ASGD would keep the **buffering logic on the workers**: each worker would accumulate local gradients and send only an *aggregated* gradient (plus the corresponding count) once enough gradients have been computed to satisfy the Phase~1 condition (i.e., when every worker has contributed at least one gradient for the current round). The server-side algorithm remains exactly as in the paper, but communication is now amortized over many local computations per worker.
> >
> > By contrast, IA-SGD necessarily communicates **every time** a worker computes a stochastic gradient, because each incoming gradient is used immediately in a model update. It cannot aggregate multiple local gradients before sending them without changing the algorithm itself.
> >
> > Therefore, under any model where each message has a non-negligible communication cost, Ringleader ASGD can be implemented to use **strictly fewer communication rounds** than IA-SGD:
> > - Ringleader ASGD can send *batched/aggregated* gradients per worker per round,
> > - IA-SGD must send *every single* gradient separately.
> >
> > In this sense, Ringleader ASGD is inherently *more communication-efficient* than IA-SGD when both are implemented in a reasonable communication-conscious way. A full formal analysis in a joint computation–communication model is beyond the scope of the current paper, but IA-SGD will not dominate Ringleader ASGD when communication costs are taken into account.

---

> > > ### Author Response · Authors · 2025-11-20
> > > **Addressing the Questions 2**
> > >
> > > > 2. It would also be good to discuss as well as implement a variant of minibatch SGD with different mini-batch sizes on each machine, which is another natural synchronous baseline in the face of different compute speeds. In a sense, this is the simplest synchronous algorithm, not the naive mini-batch discussed in Section 3. When communication costs are high, I would expect this algorithm to outperform both IA-SGD and ringleader SGD, as it requires fewer locks (a single lock per communication round between all devices). In the setup discussed in the previous bullet point, as the authors vary the communication cost, this should intuitively beat all baselines at some point.
> > >
> > > The minibatch SGD variant with different fixed mini-batch sizes on each machine that you suggest is, in general, **not implementable in a robust way**. This is especially true when client compute times are unknown, random, or even change arbitrarily over time. In such cases, it is not clear how to choose “optimal” or even reasonable per-machine batch sizes a priori.
> > >
> > > In our setting, the best one can do on top of Naive Minibatch SGD is essentially what Malenia SGD already does: it is the optimal synchronous method we discuss in Section 3. With a minor implementation change, Malenia SGD can also be made communication-efficient, in exactly the same spirit as we described above for Ringleader: workers collect/buffer gradients locally and only communicate once enough has been accumulated. In Malenia SGD, the effective batch sizes $b_i^k$ automatically reflect how fast each worker is, without having to fix them in advance. The synchronization point can be chosen as “each worker has computed at least one gradient”, or more generally, when the Malenia SGD condition (4) is satisfied. For the server to detect this, some communication is necessary, but it can be minimal (e.g., sending a single integer with the number of gradients computed, or a single signal per worker).
> > >
> > > We agree that such a synchronous scheme would have very low communication overhead. In comparison, Ringleader ASGD performs $n$ updates per round in Phase 2 and thus would incur more communication if implemented naïvely in a setting where each update involves a costly message. This highlights why extending the theory to joint computation–communication models is important: with tools such as gradient compression or local training, it may be possible to design asynchronous SGD methods that are optimal also when communication time is non-negligible. However, this is outside the scope of our current work. Addressing it would require new lower bounds, and at present it is not even clear what the “ideal” synchronous algorithm should look like in such a communication-aware model. Is it going to be the Malenia SGD that we described above? Most likely not.
> > >
> > > > The last two points highlight the age-old heuristic in distributed optimization: "Is the precious compute on the server better spent in compressing/implementing asynchrony/de-biasing or running more synchronous updates for a fixed time budget?" Naturally, this is an empirical question, and the role of the theory should be to help weed out different regimes where different choices make sense.
> > >
> > > We completely agree with this broader perspective and with the underlying trade-off you describe. Our work does **not** attempt to solve this full design problem. Instead, its goal is more focused: to understand, in a clean theoretical model, whether asynchrony is in fact the “right tool” for the specific issue it is meant to address (waiting due to heterogeneous/straggling workers), and whether it can do so optimally.
> > >
> > > In that sense, our contribution should be viewed as isolating and resolving one axis of the overall design space: we show that, in the standard computation-only model and under heterogeneous data, there exists an asynchronous method that matches the optimal time complexity of the best parallel first-order methods. We see this as a necessary step before tackling the more general—and more complex—question of how to jointly trade off asynchrony, compression, and de-biasing.
> > >
> > > We agree that identifying the regimes where different combinations (e.g., synchronous + compression vs. asynchronous + compression) are preferable, both theoretically and empirically, is an important direction for future work. Our results are intended as one building block for that broader picture, rather than a complete answer to it.

---

> > > > ### Author Response · Authors · 2025-11-20
> > > > **Addressing the Questions 3**
> > > >
> > > > > 3. The discussion around Assumption 2 and the need for data similarity assumptions in this paper is a bit unclear to me...
> > > >
> > > > Thank you for these insightful remarks and for pointing out the related literature.
> > > >
> > > > We agree that in the “non-trivial” regime where $L^2 < \frac{1}{n}\sum_i L_{f_i}^2$, one can view $L$ as implicitly reflecting both a smoothness effect and a heterogeneity effect, and that it would be very interesting to make this explicit via a second-order heterogeneity parameter (or something similar) as in the works you cited. Doing this properly would require re-deriving upper *and* lower bounds in a framework where the complexity decouples into a smoothness term and a separate heterogeneity term, which is beyond the scope of the present paper.

---

### Meta-Review · Area_Chair_pxiE · 2026-01-07

**Summary:**

This paper studies distributed nonconvex optimization with heterogeneous data across workers and heterogeneous compute times under a fixed computation-time (under a  ``computation-only”) model. It proposes Ringleader ASGD, an asynchronous method built around a gradient table (so every update uses information from all workers,even if stale), and also uses a two-phase round structure that buffers fast workers’s  extra gradients and performs exactly one update per worker per round. The goal is the is not “better rates than the optimal synchronous method,” but rather showing the existence of an asynchronous method that matches the known parallel lower bound in the heterogeneous-data setting withoutthe heterogeneity penalties.

**Reviewer Concerns:**

Reviewer rk1S :

(1) calls it “unreasonable to ignore communication costs” and asks for a dedicated section discussing limitations, known comm-aware results, and ideally experiments simulating communication costs. -- This was largely addressed by the authors by clarifying the computation-only model and explicitly state that “optimality” is relative to the standard computation-only model used in prior async-SGD theory and in the lower bound they match. On the “IA-SGD might win when comm is expensive” intuition: the authors disagree, since a communication-conscious implementation of Ringleader would batch on workers and communicate aggregated gradients.
(2) argues Assumption 2 “essentially amounts to a data heterogeneity assumption” in some regimes and asks for discussion of second-order heterogeneity literature and whether the analysis can be extended with a decoupled heterogeneity parameter.---the authors argue that the second-order heterogeneity is similar only in spirit, because Ringleader updates average across all workers using a gradient table and thus avoid an explicit heterogeneity penalty constant in the rate, and rather use a smoothness based analysis. Regarding explicitly “decouple smoothness vs heterogeneity parameter” ask, they say doing it would require re-deriving upper/lower bounds in a different framework and is beyond scope.
(3) suggests testing a synchronous minibatch method with different per-machine minibatch sizes and more generally studying regimes where “server compute” is better spent on sync vs async.--- the authors argue argue per-worker fixed minibatch sizing is not robust under unknown/time-varying compute times, and claim Malenia SGD (which they compare against in the paper) already captures the best synchronous approach

Reviewer kQKe:
(1) failure sensitivity : kQKe argues a single worker failure halts the process, so it’s a hybrid sync/async method. --- authors argue permanent worker failure makes the objective ill-posed to solve to arbitrary stationarity (since missing component cannot be recovered), but temporary slowdowns don’t halt the process; and they give a definition of async vs sync based on whether updates can use stale gradients, claiming Ringleader is async with bounded delays despite round-level coordination.
(2) Requests computational complexity analysis vs others --- authors argue time and gradient-count complexity are equivalent up to constants because all methods keep workers busy, then provide an explicit gradient-count comparison showing Ringleader has strictly better guarantees than IA-SGD and similar complexity to Malenia up to constants
(3) Minor comments about writing structure -- addressed.
(4) Lack of experiments -- not addressed, authors argue this is a theoretical work.

Reviewer M5AX
Lack of experiments -- authors argue the theoretical contributions of the paper is their main focus.

Rev nJYq
(1) No theoretical improvement possible over Malenia SGD in the same regime; asks if any regime shows advantage (e.g., with similarity assumptions) --- authors argue out of scope
(2) Impact feels limited given no theoretical advantage + small experiments; suggests the parameter-free aspect may be the only non-incremental contribution. --- authors disagree and emphasize the key contribution is showing an asynchronous method can match the optimal time complexity in the heterogeneous-data + heterogeneous compute-time model.
(3) Minor presentation issues -- addressed.

**Reviewer Scores:**

rk1S’s core objection on relationship with the data-heterogenous works and comm-aware realism or at least comm-cost experiments,largely remains unresolved. The authors insist on theoretical nature of the paper; I do not believe rk1S would have raised their score.

kQKe explicitly stated they will increase their score 4 -> 6

nJYq noted 6, remained skeptical of  the contributions. In light of other discussions, I dont beleive they would have increased their scores, could have decreased it given rk1S’s concerns which match theirs

---

### Decision · Program_Chairs · 2026-01-26

Accept (Poster)